# Nucleotide Analogues Bearing a C2′ or C3′-Stereogenic All-Carbon Quaternary Center as SARS-CoV-2 RdRp Inhibitors [note 1]

**DOI:** 10.3390/molecules27020564

**Published:** 2022-01-17

**Authors:** Amarender Manchoju, Renaud Zelli, Gang Wang, Carla Eymard, Adrian Oo, Mona Nemer, Michel Prévost, Baek Kim, Yvan Guindon

**Affiliations:** 1Bio-Organic Chemistry Laboratory, Institut de Recherches Cliniques de Montréal, Montréal, QC H2W 1R7, Canada; amarender947@gmail.com (A.M.); renaud.zelli@gmail.com (R.Z.); alanwangsherbrooke@gmail.com (G.W.); carla.eymard@ircm.qc.ca (C.E.); 2Department of Pediatrics, School of Medicine, Emory University, Atlanta, GA 30322, USA; adrian.oo@emory.edu; 3Department of Biochemistry, Microbiology and Immunology, University of Ottawa, Ottawa, ON K1N 6N5, Canada; 4Children’s Healthcare of Atlanta, Atlanta, GA 30322, USA; 5Department of Chemistry, Université de Montréal, Montréal, QC H3C 3J7, Canada

**Keywords:** SARS-CoV-2, COVID-19, RdRp, quaternary stereocenter, nucleoside analogues, epoxidation, glycosylation, triphosphorylation

## Abstract

The design of novel nucleoside triphosphate (NTP) analogues bearing an all-carbon quaternary center at C2′ or C3′ is described. The construction of this all-carbon stereogenic center involves the use of an intramoleculer photoredox-catalyzed reaction. The nucleoside analogues (NA) hydroxyl functional group at C2′ was generated by diastereoselective epoxidation. In addition, highly enantioselective and diastereoselective Mukaiyama aldol reactions, diastereoselective *N*-glycosylations and regioselective triphosphorylation reactions were employed to synthesize the novel NTPs. Two of these compounds are inhibitors of the RNA-dependent RNA polymerase (RdRp) of SARS-CoV-2, the causal virus of COVID-19.

## 1. Introduction

Severe acute respiratory syndrome coronavirus 2 (SARS-CoV-2) is a positive-sense RNA virus and the causal agent of coronavirus (CoV) disease 2019 (COVID-19). CoVs employ a multi-subunit replication/transcription machinery. The virus enters the cell by endocytosis using the ACE 2 receptors and is then uncoated. ORF1a and ORF2 of the positive strand RNA are then translated to produce non-structural protein precursors (nsp), including proteins that further cleave the precursor to form mature functional helicase and RNA-dependent RNA polymerase [1] (RdRp, or alternatively nsp 12). The latter has been recognized as an optimal target for drug design, due to its crucial role in RNA synthesis, lack of host homologues and high structural conservation between coronaviruses.

The severity of SARS-CoV-2 disease has encouraged many laboratories to evaluate potential inhibitors of RdRp from related viruses. Sofosbuvir, a potent hepatitis c virus (HCV) polymerase inhibitor, was first studied. Inhibitors of single-stranded negative RNA viruses, such as Remdesivir (Figure 1A, Ebola) [2] and B-D-N_4_- hydroxy-cytidine NHC [3] (Molnupiravir or EIDD-1931, influenza), along with other nucleotides, were also examined [4].

Remdesivir (RDV) was approved by the Food and Drug Administration (FDA) for the intravenous treatment of COVID-19 in hospitalized adult and pediatric patients [5]. RDV-TP (**1**), the active agent, is incorporated into RNA by the RdRp enzyme and then acts as a delayed chain termination of RNA synthesis at position i + 3 [6]. The Remdesivir-induced RdRp stalling is caused by a translocation barrier between the C1′-cyano group in the Remdesivir ribose moiety and the serine 861 side chain in the nsp 12 subunit of RdRp [7]. The truncation of serine 861 to alanine or glycine renders the RdRp less sensitive or insensitive to inhibition by Remdesivir [8]. An alternative mechanism was recently proposed involving the RdRp-dependent RNA proofreading. The mechanism of NHC (Molnupiravir), presently evaluated in clinical trials and is attributed to the triple level incorporation of NHC-TP, resulting in increased mutations and, ultimately, in a process known as “lethal mutagenesis”.

From a drug design standpoint, the antiviral activity of nucleoside analogues (NAs) depends on their efficient transport into cells where a first phosphorylation takes place (e.g., by cytidine kinase). These two critical steps are rate-limiting in the entire cascade that leads, through subsequent phosphorylation by other kinases, to the active nucleoside triphosphate (NTP) analogues that compete with natural nucleoside triphosphates. To circumvent these limitations, phosphoramidates were developed as monophosphate lipophilic pro-drugs that facilitate intracellular transport. The pro-drug is cleaved intracellularly, releasing the NA monophosphate that is then transformed into the corresponding bioactive triphosphorylated analogue. The main objective of the present study was to evaluate a novel series of triphosphorylated nucleotide analogues to test their activity directly in vitro against RdRp [9]. If active, the corresponding phosphoramidate pro-drugs would then be installed on the parent NA for further evaluations. The nucleoside analogues reported in the literature thus far display additional substituents at C1′ (cyano for Remdesivir Figure 1A) and different C2′ or C3′ substituents (methyl or fluorine, Figure 1B). This was suggestive that further modification of the C2′or C3′ positions would be tolerated.

Our laboratory has a long-standing interest in the development of new methodologies to improve the synthesis of nucleoside analogues. In parallel, we have been studying carbon-centered free radicals on acyclic molecules and their reactivity in atom transfer reactions, leading to the generation of all carbon stereogenic quaternary centers. Together, these findings [10,11,12,13,14,15,16] led to the conceptualization and synthesis of novel nucleoside analogues bearing a quaternary all-carbon stereogenic center at C3′ or C2′ [16,17,18].

The presence of these all-carbon quaternary centers also provides structural properties that may influence the recognition of these nucleosides or nucleotides by given targets (enzymes or a receptors). For instance, nucleosides and nucleotides are flexible molecules adopting conformations ranging between North (C3′ endo, Figure 1C) or South (C2′ endo, Figure 1C). Increasing the populations of nucleosides in their bio-active conformation may translate into a greater binding affinity to the target. We hypothesized that the presence of the quaternary center could induce a conformational bias when located at C2′ or C3′, the former would favor the North conformation (RNA-like) and the latter, the South (DNA-like). These conformational changes would be induced to minimize the steric effects (gauche effects) of the quaternary centers with their proximal substituents (Figure 1C). The X-ray analyses of some of our analogues having stereogenic quaternary centers at C2′ or C3′ supports these conformational biases. In solution, these molecules will, however, still possess some plasticity, allowing for conformational realignment during binding, contrary to locked nucleosides analogues [19]. The presence of the hydroxyl on this center could also potentially act as an extended pharmacophore, providing different proximal binding. On the other hand, binding to enzymes susceptible to steric hindrance at these positions could lead to inactivity. We thus embarked on the syntheses of a small library of nucleotides having stereogenic centers at C2′ or C3′ bearing adenine or cytosine nucleobases (Figure 1D) to investigate their inhibitory profile against RdRp.

## 2. Results and Discussion

The synthesis of the all-carbon quaternary stereogenic centers was based on our findings that a carbon-centered free radical flanked by an ester and a secondary carbon bearing an electronegative substituent (hydroxyl) could stereoselectively participate in kinetically controlled atom transfer reactions. Hydrogen and allylation transfer reactions have previously been studied both experimentally and theoretically by our group [20,21]. We recently prepared radical precursor **2** using an enantioselective Mukaiyama aldol reaction in good yield and with high diastereoselectivity in favor of the 3*R* isomer (Figure 2) [14]. The secondary alcohol was then transformed into dimethylallysilyl ether **3**. Acyclic intermediate **6,** bearing the quaternary stereogenic center, was synthesized by a sequence involving an intramolecular atom transfer cyclization, an elimination reaction under photoredox catalysis and subsequent ester reduction with protection of alcohol generated. 

Allylic oxidation of **6** using SeO_2_ led to a modest 50% yield of the corresponding ketone **7** (Figure 2) [14]. The 2,4-*syn* diol **8** was then obtained from ketone **7** by reduction using catecholborane in the presence of cesium chloride. The key intermediate **8** was transformed to furanosides **9** and **10**. The corresponding *β*-cytosine nucleoside analogue **11** was obtained from **9** by taking advantage of anchimeric participation by the acetate at C2′. On the other hand, *α*-cytosine NA **12** was accessed from **10** using Me_2_BBr [22].

Biological evaluation of the corresponding novel nucleotides necessitated an improvement in the overall synthesis of these C3′ quaternary substituted nucleosides. As reported herein (Figure 2), this was accomplished by introducing the hydroxyl at C2′ through a stereoselective epoxidation of glycal **13** [23,24] to the epoxide **14**, followed by hydrolysis and stereoselective *N*-glycosylation to generate novel nucleoside analogues **16**.

Synthesis of the requisite glycal intermediate began with secondary hydroxyl protection of methyl ester **5** with triethylsilyl ether (TES). Ester **17** was then reduced by DIBAL-H to the primary alcohol and further acetylated to give **18** in excellent yield (Figure 1). Ozonolysis of the terminal alkene **18** provided aldehyde **19** in 85% yield. 

Different pathways to prepare key intermediate **21** were explored. Cyclization of aldehyde **19** in the presence of PTSA in THF/H_2_O led to lactols **20a,b** with a 1.4:1 anomeric ratio (Figure 2). Mesylation and elimination with Et_3_N generated glycal **21** in 51% yield. Alternatively, methylfuranoside **22a,b** was derived from aldehyde **19** in the presence of PTSA in anhydrous methanol. Methylfuranoside **22a,b** were then treated with TMSOTf and 2,6-lutidine to give glycal **21** in an excellent yield.

Dondoni’s dimethyldioxirane (DMDO) [25], generated in situ with a catalytic amount of acetone and a stoichiometric amount of potassium peroxymonosulfate (oxone), was chosen as the oxidant. When glycal **21** was subjected to DMDO oxidation, the ribo-like epoxide **23a** was obtained as the major epoxide in a 7:1 ratio relative to the arabino-like epoxide **23b** (Figure 3). The stereoselectivity is rationalized by spiro-like transition states **TS A** and **TS B** (Figure 3), where DMDO approaches the olefin from the inside face of the glycals’ envelope-like conformations [26,27,28]. Attack from the bottom face in **TS A** avoids significant steric clash with the C5′ substituent in **TS B** (Figure 3). Hydrolysis of the crude epoxides **23a** and **23b** in THF/H_2_O gave the ribo-like lactols **24a,b** as the major compounds (63%, isolated by flash column chromatography). These lactols were then protected with acetyl groups to afford acetate ribofuranosides **25a,b** in 85% yield. 

With ribo-like diacetate furanosides **25a**,**b** in hand, stereoselective *N*-glycosylations of **25****a**,**b** were performed in the presence of a silylated base (adenine or cytidine) and TMSOTf (Figure 4). The 1′,2′-*trans* ribo-like nucleosides **26**–**28** were obtained in excellent diastereoselectivity (>20:1 dr) and yield (75–85%) in accordance with anchimeric assistance of the C2′ acetyl group. NMR spectroscopic analysis experiments confirmed the 1,2′-*trans* stereochemistries (2D NOESY) and, in the case of adenine coupling, N^9^ isomers (HMBC). Subsequent cleavage of the C5′-OTBDPS protecting groups with 3HF·NEt_3_ provided corresponding nucleoside analogues **29**–**31** (86–87%), which were the key precursors for the formation of the nucleoside C5′O-triphosphates. Further, cleavage of the C2′and C3′ acetyl-protecting groups with NaOMe provided 1′,2′-*trans* ribo-like nucleoside analogues **32**–**34** (80–82%). 

The synthesis of the C2′ quaternary series was then explored. The five-membered ring lactone **40,** bearing the quaternary center at C2′, was prepared by a route previously reported by our group [13] (Figure 5). The Mukaiyama aldol reaction of aldehyde **35** with tetrasubstituted silylated-enolether **36** provided **37a**,**b**. The α-bromomethylesters **37a**,**b** were subjected to lactonization, followed by installation of a TBS at the C5 primary alcohol. Installation of vinyldimethylsilane at the C3′ secondary alcohol provided lactones **39a,b,** which were subjected to an atom transfer cyclization/elimination reaction using photoredox catalysis to give lactone **40**. An alternative synthetic sequence was then optimized to reduce the number of steps to reach **43a**,**b** [13]. A TBS-protecting group was introduced on the secondary hydroxyl of **40**, followed by ozonolysis resulting in aldehyde **41** in excellent yield. After the simultaneous reduction of both the lactone and aldehyde using Red-Al, benzoylation provided furanosides **43a**,**b**.

With benzoylated furanosides **43a**,**b** in hand, stereoselective *N*-glycosidation of **43a**,**b** with 2,6 dichloropurine in the presence of TMSOTf at −10 °C led to the regioselective formation of N^9^ nucleoside analogue **44** with high diastereoselectivity (*β*:*α*, 10:1) and good yield (73%, Figure 6). The selective formation of the *β*-anomer is attributed to an anchimeric assistance from C2′ benzoate. The 1′,2′-*trans* stereochemistry was confirmed by 2D NOESY experiments, while the N^9^ regioselectivity was verified by the key indicative three bond correlation between the H1′ of sugar and C4 of purine in ^1^H/^13^C 2D HMBC NMR experiments. Nucleoside **45** was formed from treatment with ammonia in methanol to give the 2-chloroadenosine and deprotection of the C2′ benzoate, followed by desilylation in the presence of 3HF·NEt_3_. The 2-chloroadenosine analogue **45** was then hydrogenated to the adenosine derivative **46** in 78% yield. The cytidine analogue **47** was also prepared from the benzoylated furanoside, as reported by our group [13].

### 2.1. Synthesis of Nucleoside Triphosphates (NTPs)

Having established synthetic routes to access nucleoside analogues bearing an all-carbon quaternary stereocenter at C2′ or C3′, we next investigated the synthesis of their C5′ triphosphate derivatives. The synthesis of NTPs represents a challenging task [29]. Initial triphosphorylation attempts using Taylor’s method, using trimetaphosphate and mesitylenesulfonyl chloride, were unsuccessful [30]. The one-pot synthesis approach of Huang and co-workers [31] was more appropriate for our substrates. The phosphorylation is accomplished under mild conditions using tributylammonium pyrophosphate in the presence of salicyl phosphorochloridite (SalPCl).

Phosphorylation of 1′,2′-*trans* ribo-like nucleoside analogues bearing either purine or pyrimidine nucleobases was carried out using SalPCl, (Bu_2_HN)_2_H_2_P_2_O_7_ and Bu_3_N in anhydrous DMF. Subsequent addition of protected nucleoside provided the five-cyclic triphosphite intermediates that were then subjected to iodine oxidation and hydrolysis. The cleavage of the C2′ and C3′acetyl-protecting groups with ammonium hydroxide (NH_4_OH) then generated the corresponding 1′,2′-*trans* ribo-like nucleoside 5′-triphosphate (Figure 7). The final 1′,2′-*trans* ribo-like NTPs (**2**–**4**) were prepared in good yields (27–51%, Figure 7) from the nucleoside analogues **29**–**31**, respectively.

The phosphorylation of nucleoside analogues bearing a C2′ quaternary stereocenter was then conducted using the same strategy. Triphosphorylation of unprotected NAs **46**–**47** furnished the corresponding NTPs **5**–**7** in low, but acceptable yields, for this challenging transformation (5–13%, Figure 8).

### 2.2. SARS-CoV-2 RNA-Dependent RNA Polymerase (RdRp) Assay

Preliminary results for SARS-CoV-2 RNA-dependent RNA polymerase (RdRp) inhibition suggested that the most active analogues were LCB-2344 (**6**) and LCB-2279 (**7**), which both bear purine nucleobases and the quaternary stereogenic center at C2′ (Figure 3). These NTPs (**6** and **7**) appear to have just slightly lower activity than the commercial Remdesivir TP (**1**). We also prepared the REM-TP (1) using the approach described at Figure 8, the activity against RdRp obtained with this product was similar to 6 (LCB-2344). NTP **5**, bearing a cytosine nucleobase, was inactive. Nucleotide analogues (**2**–**4**), having the adenine, chloro-adenine and cytosine nucleobases at C1′ and the quaternary center at C3′, showed a lower activity profile.

## 3. Materials and Methods

### 3.1. General Information

All the anhydrous solvents were purchased from Sigma Aldrich, Saint Louis, MO, USA. All the glassware was purchased from Pyrex USA. Reactions requiring anhydrous conditions were performed under an atmosphere of nitrogen in flame-dried glassware using standard syringe techniques. Molecular sieves (Sigma Aldrich, USA) were used to prepare the anhydrous solvents. The sieves (4 Å, 1–2 mm beads) were activated by heating at 180 °C for 48 h under vacuum prior to their addition into new bottles of solvent purged with argon. All commercially available reagents were used as received: TBSCl was purchased from TCI America, USA and TBSOTf was obtained from Oakwood Chemicals, USA. Remaining all reagents were purchased from Sigma Aldrich, USA. Flash chromatography was done on silica gel 60 (0.040–0.063 mm, Silicycle, QC, Canada) using an automated purification system. Thin-layer chromatography (TLC) was done on pre-coated (0.25 mm) Merck F-254 silica gel aluminum plates. Visualization was performed with short UV wavelengths and/or revealed with potassium permanganate solutions. See the ^1^H, ^13^C, and 2D NMR spectra data in the Appendix A. The ^1^H NMR spectra were recorded at room temperature on a 700 MHz, 500 MHz and 400 MHz NMR spectrometer and the data is reported as follows: chemical shift in ppm referenced to residual solvent (CDCl_3_ δ 7.26, CD_3_OD δ 3.31 and D_2_O δ 4.79 ppm), multiplicity (s = singlet, d = doublet, dd = doublet of doublets, ddd = doublet of doublets of doublets, t = triplet, td = triplet of doublets, m = multiplet, app = apparent), coupling constants (Hz), and integration. The ^13^C NMR spectra were recorded at room temperature using 176 MHz and 126 MHz with the data reported as follows: chemical shift in ppm referenced to residual solvent (CDCl_3_ δ 77.16 and CD_3_OD δ 49.00 ppm). The ^31^P NMR spectra were recorded at room temperature using 162 MHz. Infrared spectra were recorded on a Fourier transform infrared spectrophotometer from a thin film of purified product or with single reflection diamond attenuated total reflection module, and signals are reported in cm^−1^. Mass spectra were recorded through electrospray ionization with positive ion mode. A Hybrid Quadrupole Orbitrap mass analyzer was used for high-resolution mass spectrometry (HRMS) measurements. Optical rotations were measured at room temperature from the sodium D line (589 nm) using the formula: [α]_D_ = (100)α_obs_//(ℓ·(c)), where c = (g of substrate/100 mL of solvent) and ℓ = 1 dm. The sequence from **35** to **40** in the Figure 5, and compounds **5** and **47** [13] were prepared as reported in our previous publications [13,14]. 

### 3.2. Synthesis

*(*−*)-Methyl (S)-2-((S)-3,3-diethyl-9,9-dimethyl-8,8-diphenyl-4,7-dioxa-3,8-disiladecan-5-yl)-2-methylpent-4-enoate* (**17**). To a solution of secondary alcohol **5** [14] (3.90 g, 9.14 mmol, 1.00 equiv.) in anhydrous DCM (45 mL, 0.20 M), imidazole (1.56 g, 22.9 mmol, 2.50 equiv.) was added, immediately followed by triethylchlorosilane (2.30 mL, 13.7 mmol, 1.50 equiv.). The resulting mixture was stirred at room temperature for 16 h. The mixture was diluted with DCM (25 mL) and saturated NH_4_Cl solution (25 mL). The aqueous phase was extracted with DCM (3 × 25 mL) and the combined organic phases were washed with brine, dried over MgSO_4_, filtered and concentrated under reduced pressure. Purification by flash chromatography on silica gel (Hexanes/EtOAc, 97:3) provided the TES-protected methyl ester **17** (4.3 g, 87%) as a colorless oil. R*_f_* = 0.43 (Hexanes/EtOAc, 95:5); [α]^25^_D_ −27 (*c* 0.7, CH_2_Cl_2_); Formula: C_31_H_48_O_4_Si_2_; MW: 540.82 g/mol; IR (neat) ν_max_ 3073, 3051, 2954, 2879, 1738 cm^−1^; ^1^H NMR (500 MHz, CDCl_3_) δ 7.66–7.63 (m, 4H), 7.44–7.36 (m, 6H), 5.71–5.63 (m, 1H), 5.02 (s, 1H), 5.01–4.98 (m, 1H), 4.15 (t, *J* = 5.9 Hz, 1H), 3.52 (d, *J* = 5.9 Hz, 2H), 3.46 (s, 3H), 2.39 (dd, *J* = 13.6, 7.3 Hz, 1H), 2.25 (dd, *J* = 13.3, 7.7 Hz, 1H), 1.06 (s, 3H), 1.04 (s, 9H), 0.89 (t, *J* = 7.9 Hz, 9H), 0.56 (q, *J* = 8.0 Hz, 6H) ppm; ^13^C NMR (126 MHz, CDCl_3_) δ 175.5, 135.83 (2C), 135.78 (2C), 134.3, 133.5, 133.3, 129.80 (2C), 129.76 (2C), 127.80, 127.75, 118.0, 77.3, 66.4, 51.5, 50.6, 42.1, 27.0 (3C), 19.3, 14.6, 7.0 (3C), 5.3 (3C) ppm; HRMS (ESI) *m/z* [M + Na]^+^ calcd for C_31_H_48_O_4_Si_2_Na: 563.2983; found 563.2985 (+0.4 ppm).

*(−)-(R)-2-((S)-3,3-Diethyl-9,9-dimethyl-8,8-diphenyl-4,7-dioxa-3,8-disiladecan-5-yl)-2-methylpent-4-en-1-ol* (**S1**). To a solution of methyl ester **17** (4.30 g, 7.95 mmol, 1.00 equiv.) in anhydrous DCM (40 mL, 0.20 M) at −40 °C, DIBAL-H (20 mL, 20 mmol, 2.5 equiv., 1.0 M in hexanes) was added dropwise. The resulting mixture was stirred at −40 °C for 2 h. Methanol (1.3 mL, 32 mmol, 4.0 equiv.) was added at −40 °C and the reaction was stirred for 10 min followed by addition of Et_2_O (50 mL) and saturated Rochelle salt solution (50 mL). The solution was stirred vigorously until separation of both layers. The aqueous phase was extracted with Et_2_O (2 × 50 mL), and the combined organic layers were washed with water, dried over MgSO_4_, filtered and concentrated under reduced pressure. The ^1^H NMR analysis indicated that alcohol **S1** (4.08 g, quant., colorless oil) was clean enough to be directly used in the next step without further purification. R*_f_* = 0.45 (Hexanes/EtOAc, 90:10); [α]^25^_D_ −7.4 (*c* 1.1, CH_2_Cl_2_); Formula: C_30_H_48_O_3_Si_2_; MW: 512.88 g/mol; IR (neat) ν_max_ 3461 (br), 3072, 3051, 2955, 2934, 2877 cm^−1^; ^1^H NMR (500 MHz, CDCl_3_) δ 7.68–7.66 (m, 4H), 7.46–7.39 (m, 6H), 5.83–5.74 (m, 1H), 5.03–4.97 (m, 2H), 3.78 (dd, *J* = 10.9, 5.8 Hz, 1H), 3.70 (dd, *J* = 5.5, 4.4 Hz, 1H), 3.56 (dd, *J* = 10.9, 4.3 Hz, 1H), 3.52 (dd, *J* = 11.5, 6.3 Hz, 1H), 3.44 (dd, *J* = 11.4, 6.2 Hz, 1H), 3.18 (t, *J* = 6.3 Hz, 1H), 2.14 (dd, *J* = 13.6, 7.4 Hz, 1H), 1.97 (dd, *J* = 13.6, 7.5 Hz, 1H), 1.07 (s, 9H), 0.87 (t, *J* = 8.0 Hz, 9H), 0.85 (s, 3H), 0.52 (q, *J* = 7.9 Hz, 6H) ppm; ^13^C NMR (126 MHz, CDCl_3_) δ 135.90 (2C), 135.85 (2C), 134.7, 132.94, 132.87, 130.04, 130.02, 127.92 (2C), 127.91 (2C), 117.7, 79.6, 68.1, 66.0, 42.1, 39.1, 27.0 (3C), 19.2, 18.4, 7.0 (3C), 5.1 (3C) ppm; HRMS (ESI) *m/z* [M + Na]^+^ calcd for C_30_H_48_O_3_Si_2_Na: 535.3034; found 535.3042 (+1.4 ppm).

*(+)-(R)-2-((S)-3,3-Diethyl-9,9-dimethyl-8,8-diphenyl-4,7-dioxa-3,8-disiladecan-5-yl)-2-methylpent-4-en-1-yl acetate* (**18**). To a solution of primary alcohol **S1** (2.25 g, 4.39 mmol, 1.00 equiv.), a 1:2 mixture of pyridine and Ac_2_O (18 mL, 0.25 M) was added at 0 °C and the resulting mixture was stirred at room temperature for 1 h. The solution was concentrated under reduced pressure and the crude product was purified by silica gel flash chromatography (Hexanes/EtOAc, 95:5), providing the acetate product **18** as a colorless oil (2.21 g, 91% over two steps). R*_f_* = 0.65 (Hexanes/EtOAc, 90:10); [α]^25^_D_ +0.4 (*c* 0.6, CH_2_Cl_2_); Formula: C_32_H_50_O_4_Si_2_; MW: 554.92 g/mol; IR (neat) ν_max_ 3073, 3051, 2956, 2934, 2877, 2859, 1744 cm^−1^; ^1^H NMR (500 MHz, CDCl_3_) δ 7.68–7.66 (m, 4H), 7.45–7.37 (m, 6H), 5.76–5.69 (m, 1H), 5.00–4.92 (m, 2H), 3.95–3.90 (m, 2H), 3.79–3.74 (m, 2H), 3.54 (dd, *J* = 10.0, 4.7 Hz, 1H), 2.15 (dd, *J* = 13.9, 7.4 Hz, 1H), 2.01 (dd, *J* = 13.8, 7.6 Hz, 1H), 1.97 (s, 3H), 1.07 (s, 9H), 0.91 (t, *J* = 7.9 Hz, 9H), 0.83 (s, 3H), 0.62–0.54 (m, 6H) ppm; ^13^C NMR (126 MHz, CDCl_3_) δ 171.0, 135.9 (2C), 135.8 (2C), 134.5, 133.5, 133.4, 129.9, 129.8, 127.8 (4C), 117.7, 77.4, 68.3, 66.5, 41.1, 38.5, 27.0 (3C), 21.0, 19.3, 18.9, 7.1 (3C), 5.3 (3C) ppm; HRMS (ESI) *m/z* [M + H]^+^ calcd for C_32_H_51_O_4_Si_2_: 555.3320; found 555.3328 (+1.4 ppm).

*(*−*)-(2R,3S)-4-((tert-Butyldiphenylsilyl)oxy)-2-methyl-2-(2-oxoethyl)-3-((triethylsilyl)oxy) butyl acetate* (**19**). To a solution of alkene **18** (2.21 g, 3.98 mmol, 1.00 equiv.) in anhydrous DCM (60 mL, 0.070 M) at −78 °C, ozone was bubbled under vacuum until the solution turned pale blue (about 25 min). The reaction was then purged with nitrogen to remove excess ozone. After addition of Et_3_N (1.67 mL, 11.9 mmol, 3.00 equiv.), the solution was kept at −78 °C for 30 min and then warmed to room temperature for 1 h. MgSO_4_ was added, and the resulting mixture was filtered and concentrated in vacuo. The crude product was purified by silica gel flash chromatography (Hexanes/EtOAc, 90:10) to provide aldehyde **19** as a colorless oil (1.89 g, 85%). R*_f_* = 0.42 (Hexanes/EtOAc, 90:10); [α]^25^_D_ −5.5 (*c* 0.7, CH_2_Cl_2_); Formula: C_31_H_48_O_5_Si_2_; MW: 556.89 g/mol; IR (neat) ν_max_ 3072, 3050, 2955, 2935, 2877, 2859, 1745, 1720 cm^−1^; ^1^H NMR (500 MHz, CDCl_3_) δ 9.78 (t, *J* = 2.9 Hz, 1H), 7.66–7.65 (m, 4H), 7.44–7.38 (m, 6H), 4.15 (d, *J* = 11.0 Hz, 1H), 4.10 (d, *J* = 11.0 Hz, 1H), 3.79–3.74 (m, 2H), 3.56 (dd, *J* = 10.1, 3.7 Hz, 1H), 2.45 (dd, *J* = 15.3, 3.2 Hz, 1H), 2.32 (dd, *J* = 15.3, 2.6 Hz, 1H), 2.03 (s, 3H), 1.07 (s, 9H), 1.03 (s, 3H), 0.86 (t, *J* = 8.0 Hz, 9H), 0.51 (q, *J* = 7.9 Hz, 6H) ppm; ^13^C NMR (126 MHz, CDCl_3_) δ 201.6, 170.8, 135.9 (2C), 135.8 (2C), 133.12, 133.09, 130.01, 129.99, 127.9 (4C), 76.6, 68.9, 66.0, 48.5, 42.4, 27.0 (3C), 20.9, 19.6, 19.2, 7.0 (3C), 5.1 (3C) ppm; HRMS (ESI) *m/z* [M − H]^−^ calcd for C_31_H_47_O_5_Si_2_: 555.2968; found 555.2953 (−2.7 ppm).

*((2S,3R)-2-(((tert-Butyldiphenylsilyl)oxy)methyl)-5-hydroxy-3-methyltetrahydrofuran-3-yl)methyl acetate* (**20a**,**b**). Aldehyde **19** (1.88 g, 3.38 mmol, 1.00 equiv.) was dissolved in a THF and H_2_O mixture (4:1) (34 mL, 0.1 M) followed by the addition of PTSA (963 mg, 5.06 mmol, 1.50 equiv.). The resulting solution was stirred at 50 °C for 1.5 h. The solution was diluted with DCM (30 mL) and a saturated solution of NaHCO_3_ (20 mL) was added. The aqueous layer was extracted with DCM (3 × 25 mL) and the combined organic layers were dried over MgSO_4_, filtered and concentrated under reduced pressure. The residue was purified by silica gel flash chromatography (Hexanes/EtOAc, 80:20) to provide lactols **20a,b** (1.3 g, 87%, dr 1.4:1) as a colorless oil. R***_f_*** = 0.38 (Hexanes/EtOAc, 70:30); Formula: C_25_H_34_O_5_Si; MW: 442.63 g/mol; IR (neat) ν_max_ 3428 (br), 3071, 3050, 2955, 2932, 2887, 2858, 1742 cm^−1^; ^1^H NMR (500 MHz, CDCl_3_) δ 7.71–7.67 (m, 8H, major and minor), 7.46–7.37 (m, 12H, major and minor), 5.58–5.55 (m, 1H, major), 5.46–5.43 (m, 1H, minor), 4.11–4.09 (m, 3H, major), 3.92–3.85 (m, 3H, minor), 3.73–3.66 (m, 4H, major and minor), 3.23 (d, *J* = 7.5 Hz, 1H, minor), 2.74 (d, *J* = 4.3 Hz, 1H, major), 2.21 (dd, *J* = 13.7, 6.1 Hz, 1H, minor), 2.06 (dd, *J* = 13.0, 5.8 Hz, 1H, major), 2.033 (s, 3H, minor), 2.027 (s, 3H, major), 1.90 (dd, *J* = 13.6, 2.7 Hz, 1H, major), 1.81 (dd, *J* = 14.0, 3.1 Hz, 1H, minor), 1.18 (s, 3H, minor), 1.09 (s, 9H, minor), 1.06 (s, 3H, major), 1.05 (s, 9H, major) ppm; ^13^C NMR (126 MHz, CDCl_3_) δ 171.10 (minor), 171.07 (major), 135.9 (minor, 2C), 135.8 (major, 2C), 135.7 (major and minor, 4C), 133.4 (major), 133.3 (major), 132.81 (minor), 132.77 (minor), 130.1 (minor), 130.0 (minor), 129.9 (major, 2C), 127.98 (minor, 2C), 127.95 (minor, 2C), 127.86 (major, 2C), 127.84 (major, 2C), 98.7 (minor), 97.8 (major), 83.5 (minor), 82.4 (major), 70.8 (minor), 70.4 (major), 64.8 (minor), 63.9 (major), 45.3 (major), 45.2 (minor), 44.4 (minor), 44.2 (major), 27.1 (minor, 3C), 26.9 (major, 3C), 21.0 (major and minor), 19.3 (minor), 19.2 (major), 18.6 (major), 18.4 (minor) ppm; HRMS (ESI) *m/z* [M + Na]^+^ calcd for C_25_H_34_O_5_SiNa^+^: 465.2068; found 465.2070 (+0.4 ppm).

*(*+*)-((2S,3R)-2-(((tert-Butyldiphenylsilyl)oxy)methyl)-3-methyl-2,3-dihydrofuran-3-yl)methyl acetate* (**21**). To a solution of lactols **20a,b** (1.27 g, 2.87 mmol, 1.00 equiv.) in anhydrous DCE (72 mL, 0.04 M), MsCl (0.78 mL, 10 mmol, 3.5 equiv.) was added. The resulting solution was stirred 3 min at room temperature, followed by 3 min at 75 °C. Triethylamine (3.0 mL, 22 mmol, 7.5 equiv.) was then added and the resulting solution was stirred 3 min at 75 °C. After cooling to room temperature, a saturated solution of NaHCO_3_ (20 mL) was added and the aqueous layer was extracted with Et_2_O (3 × 25 mL). The combined organic layers were dried over MgSO_4_, filtered and concentrated under reduced pressure. Purification by silica gel flash chromatography (Hexanes/EtOAc, 90:10) provided glycal **21** (620 mg, 51%) as a colorless oil. R*_f_* = 0.65 (Hexanes/EtOAc, 80:20); [α]^25^_D_ +59 (*c* 0.8, CH_2_Cl_2_); Formula: C_25_H_32_O_4_Si; MW: 424.61 g/mol; IR (neat) ν_max_ 3071, 3050, 2958, 2932, 2887, 2858, 1743 cm^−1^; ^1^H NMR (500 MHz, CDCl_3_) δ 7.70–7.68 (m, 4H), 7.45–7.37 (m, 6H), 6.28 (d, *J* = 2.7 Hz, 1H), 4.74 (d, *J* = 2.7 Hz, 1H), 4.33 (t, *J* = 6.1 Hz, 1H), 4.06 (d, *J* = 10.8 Hz, 1H), 3.89 (d, *J* = 10.8 Hz, 1H), 3.87–3.80 (m, 2H), 2.02 (s, 3H), 1.08 (s, 3H), 1.07 (s, 9H) ppm; ^13^C NMR (126 MHz, CDCl_3_) δ 171.1, 145.6, 135.76 (2C), 135.75 (2C), 133.5, 133.4, 129.89, 129.87, 127.9 (4C), 107.4, 85.7, 70.9, 63.0, 48.1, 27.0 (3C), 21.0, 19.3, 18.0 ppm; HRMS (ESI) *m/z* [M + H]^+^ calcd for C_25_H_33_O_4_Si^+^: 425.2143; found 425.2148 (+1.2 ppm).

*((2S,3R)-2-(((tert-Butyldiphenylsilyl)oxy)methyl)-5-methoxy-3-methyltetrahydrofuran-3-yl)methyl acetate* (**22a**,**b**). To a solution of aldehyde **19** (1.12 g, 2.00 mmol, 1.00 equiv.) in anhydrous MeOH (10 mL, 0.20 M), PTSA (0.19 g, 1.0 mmol, 0.50 equiv.) was added at room temperature. The resulting mixture was stirred for 20 min until completion as indicated by TLC. The reaction was neutralized by addition of anhydrous Et_3_N (0.56 mL, 4.0 mmol, 2.0 equiv.) and the resulting mixture was concentrated under reduced pressure. Purification by silica gel flash chromatography (Hexanes/EtOAc, 90:10) provided methoxy lactols **22a**,**b** (0.86 g, 94%, dr 1.4:1) as a colorless oil. R*_f_* = 0.48 (Hexanes/EtOAc, 80:20); Formula: C_26_H_36_O_5_Si; MW: 456.65 g/mol; IR (neat) ν_max_ 3071, 3049, 2954, 2931, 2890, 2858, 1743 cm^−1^; ^1^H NMR (500 MHz, CDCl_3_) δ 7.69–7.68 (m, 8H, major and minor), 7.45–7.36 (m, 12H, major and minor), 5.03 (dd, *J* = 5.7, 2.9 Hz, 1H, major), 4.94 (dd, *J* = 6.0, 2.3 Hz, 1H, minor), 4.10 (d, *J* = 10.8 Hz, 1H, major), 4.00–3.93 (m, 5H, major and minor), 3.80–3.72 (m, 4H, major and minor), 3.36 (s, 3H, major), 3.30 (s, 3H, minor), 2.12 (dd, *J* = 13.5, 5.9 Hz, 1H, minor), 2.05 (s, 3H, minor), 1.99 (s, 3H, major), 1.97 (dd, *J* = 13.6, 5.8 Hz, 1H, major), 1.88 (dd, *J* = 13.6, 2.9 Hz, 1H, major), 1.74 (dd, *J* = 13.5, 2.2 Hz, 1H, minor), 1.18 (s, 3H, minor), 1.06 (s, 18H, major and minor), 1.01 (s, 3H, major) ppm; ^13^C NMR (126 MHz, CDCl_3_) δ 171.12 (minor), 171.05 (major), 135.77 (major and minor, 4C), 135.75 (minor, 2C), 135.73 (major, 2C), 133.65 (minor), 133.56 (major), 133.52 (minor), 133.47 (major), 129.9 (major), 129.84 (minor), 129.82 (major and minor), 127.8 (major and minor, 8C), 105.0 (minor), 104.2 (major), 83.6 (minor), 82.0 (major), 70.8 (minor), 70.3 (major), 64.4 (minor), 63.8 (major), 55.4 (minor), 55.2 (major), 44.5 (major), 43.9 (minor), 43.7 (minor), 43.5 (major), 26.9 (major and minor, 6C), 20.99 (minor), 20.96 (major), 19.33 (minor), 19.30 (major), 18.6 (major), 18.4 (minor) ppm; HRMS (ESI) *m/z* [M + Na]^+^ calcd for C_26_H_36_O_5_SiNa: 479.2224; found 479.2220 (−0.9 ppm).

*(+)-((2S,3R)-2-(((tert-Butyldiphenylsilyl)oxy)methyl)-3-methyl-2,3-dihydrofuran-3-yl)methyl acetate* (**21**). To a solution of methoxy lactols **22a**,**b** (0.12 g, 0.25 mmol, 1.0 equiv.) in anhydrous DCM (1.3 mL, 0.2 M), 2,6-lutidine (0.12 mL, 1.0 mmol, 4.0 equiv.) and TMSOTf (0.09 mL, 0.5 mmol, 2 equiv.) were added at 0 °C [32].The resulting solution was stirred for 30 min at room temperature. The mixture was then diluted in DCM (5 mL) followed by washing with water (5 mL), and the aqueous layer was extracted with DCM (3 × 5 mL). The combined organic layers were dried over MgSO_4_, filtered and concentrated under reduced pressure. Purification by silica gel flash chromatography (Hexanes/EtOAc, 90:10) provided glycal **21** (96 mg, 90%) as a colorless oil, which was confirmed to be identical to the compound formed from **20a**,**b** (see above). 

*((2S,3S,4R)-2-(((tert-Butyldiphenylsilyl)oxy)methyl)-4,5-dihydroxy-3-methyltetra hydrofuran-3-yl)methyl acetate* (**24a**,**b**). To a solution of glycal **21** (118 mg, 0.278 mmol, 1.00 equiv.) in anhydrous DCM (1.3 mL, 0.22 M) at 0 °C, acetone (0.13 mL, 1.6 mmol, 6.0 equiv.) and a saturated NaHCO_3_ solution (2.5 mL) were added. To the resulting biphasic mixture, a 0.37 mM solution of oxone in water (1.5 mL) was added. After sealing the flask, the resulting mixture was stirred for 30 min at 0 °C and 3 h at room temperature. After degassing the flask, the aqueous phase was extracted with DCM (3 × 5 mL). The organic layers were dried over MgSO_4_, filtered and concentrated under reduced pressure. The resulting crude epoxide **23a** and **23b** (*dr* 7:1, determined by ^1^H NMR of crude epoxide) was stirred in a THF and H_2_O (1:1) mixture (5.5 mL, 0.05 M) for 1 h. The aqueous phase was extracted with EtOAc (3 × 5 mL), and the resulting organic phase was dried over MgSO_4_, filtered and concentrated under reduced pressure. Purification by flash chromatography on silica gel (DCM/EtOAc, 80:20) provided lactols **24a**,**b** (80 mg, 63%, 4:1 mixture) as a white foam. R*_f_* = 0.33 (DCM/EtOAc, 85:15); Formula: C_25_H_34_O_6_Si; MW: 458.63 g/mol; IR (Neat) ν_max_ 3421 (br), 2931, 2857, 1741, 1720 cm^−1^; ^1^H NMR (500 MHz, CDCl_3_) δ 7.68–7.63 (m, 8H, major and minor), 7.45–7.37 (m, 12H, major and minor), 5.42 (dd, *J* = 9.6, 3.9 Hz, 1H, major), 5.23 (dd, *J* = 5.4, 1.1 Hz, 1H, minor), 4.53 (d, *J* = 11.4 Hz, 1H, major), 4.39 (d, *J* = 11.4 Hz, 1H, minor), 4.18 (d, *J* = 11.4 Hz, 1H, major), 4.06 (d, *J* = 11.4 Hz, 1H, minor), 4.11 (dd, *J* = 7.5, 4.9 Hz, 1H, minor), 4.02 (dd, *J* = 6.0, 4.9 Hz, 1H, minor), 3.89 (d, *J* = 4.2 Hz, 1H, major), 3.87 (d, *J* = 2.9 Hz, 1H, minor), 3.80–3.73 (m, 3H, major and minor), 3.64 (dd, *J* = 10.6, 7.5 Hz, 1H, major), 2.10 (s, 3H, major), 2.09 (s, 3H, minor), 1.18 (s, 3H, minor), 1.08 (s, 9H, minor), 1.05 (s, 9H, major), 1.03 (s, 3H, major) ppm; *OH signals are missing possibly due to exchange in* CDCl_3_ ppm; ^13^C NMR (126 MHz, CDCl_3_) δ 172.3 (major), 172.0 (minor), 135.74 (minor, 2C), 135.69 (minor, 2C), 135.67 (major, 2C), 135.65 (major, 2C), 133.1 (major), 133.0 (major), 132.9 (minor), 132.8 (minor), 130.04 (minor), 130.02 (minor), 129.95 (major, 2C), 128.0 (minor, 2C), 127.94 (minor, 2C), 127.90 (major, 2C), 127.89 (major, 2C), 103.9 (minor), 96.8 (major), 83.6 (minor) 83.0 (minor), 80.1 (major), 77.3 (major), 66.8 (minor), 66.6 (major), 63.9 (minor), 63.0 (major), 48.4 (major), 47.4 (minor), 27.0 (minor, 3C), 26.9 (major, 3C), 21.0 (major and minor), 19.3 (minor), 19.2 (major), 15.7 (minor), 15.4 (major) ppm; HRMS (ESI) *m/z* [M + NH_4_]^+^ calcd for C_25_H_38_O_6_NSi: 476.2463; found 476.2462 (−0.2 ppm).

(*3R,4R,5S)-4-(Acetoxymethyl)-5-(((tert-butyldiphenylsilyl)oxy)methyl)-4-methyltetra hydrofuran-2,3-diyl diacetate* (**25a**,**b**). Lactols **24a**,**b** (65 mg, 0.14 mmol) were stirred in a solution of Ac_2_O:Pyr (2:1, 1.0 mL, 0.14 M) at room temperature for 18 h and then concentrated. Purification by flash chromatography on silica gel (Hexanes/EtOAc, 60:40), provided acetates **25a**,**b** (65 mg, 85%, 4:1 mixture) as a colorless oil. R*_f_* = 0.53 (Hexanes/EtOAc, 70:30); Formula: C_29_H_38_O_8_Si; MW: 542.70 g/mol; IR (Neat) ν_max_ 2933, 2858, 1746 cm^−1^; ^1^H NMR (500 MHz, CDCl_3_) δ 7.70–7.65 (m, 8H, major and minor), 7.46–7.37 (m, 12H, major and minor), 6.41 (d, *J* = 4.8 Hz, 1H, major), 6.00 (d, *J* = 1.6 Hz, 1H, minor), 5.35 (d, *J* = 4.8 Hz, 1H, major), 5.31 (d, *J* = 1.6 Hz, 1H, minor), 4.29 (d, *J* = 11.2 Hz, 1H, major), 4.25–4.23 (m, 2H, major and minor), 4.24 (d, *J* = 11.2 Hz, 1H, major), 4.16–4.10 (m, 2H, minor), 3.77–3.74 (m, 3H, major and minor), 3.68 (dd, *J* = 11.3, 3.5 Hz, 1H, major), 2.11 (s, 3H, minor), 2.10 (s, 3H, major), 2.07 (s, 3H, major), 2.03 (s, 3H, minor), 2.03 (s, 3H, major), 1.98 (s, 3H, minor), 1.28 (s, 3H, minor), 1.25 (s, 3H, major), 1.08 (s, 9H, minor), 1.08 (s, 9H, major) ppm; ^13^C NMR (126 MHz, CDCl_3_) δ 170.9 (major), 170.7 (minor), 169.8 (major), 169.7 (minor), 169.64 (minor), 169.61 (major), 135.8 (major, 2C), 135.72 (major, 2C), 135.71 (minor, 2C), 135.69 (minor, 2C), 133.03 (minor), 132.97 (major), 132.8 (minor), 132.7 (major), 130.01 (minor), 129.98 (major, 2C), 129.95 (minor), 127.94 (major, 2C), 127.92 (major and minor, 4C), 127.89 (minor, 2C), 100.0 (minor), 94.2 (major), 84.9 (minor), 82.9 (major), 82.0 (minor), 77.9 (major), 66.8 (major), 66.4 (minor), 63.4 (major), 63.2 (minor), 46.1 (minor), 45.2 (major), 26.90 (minor, 3C), 26.88 (major, 3C), 21.21 (minor), 21.19 (major), 20.93 (major), 20.89 (minor), 20.85 (minor), 20.6 (major), 19.3 (minor), 19.2 (major), 16.8 (major), 15.8 (minor) ppm; HRMS (ESI) *m/z* [M + NH_4_]^+^ calcd for C_29_H_42_O_8_NSi: 560.2674; found 560.2673 (−0.2 ppm).

*(+)-(2R,3R,4R,5S)-2-(4-Acetamido-2-oxopyrimidin-1(2H)-yl)-4-(acetoxymethyl)-5-(((tert-butyldiphenylsilyl)oxy)methyl)-4-methyltetrahydrofuran-3-yl acetate* (**26**). To a solution of acetates **25a**,**b** (70 mg, 0.13 mmol, 1.0 equiv.) in anhydrous MeCN (0.6 mL, 0.2 M), silylated *N*^4^-acetylcytosine (0.52 mL, 0.21 mmol, 1.6 equiv. 0.40 M in DCE) was added at room temperature. The resulting mixture was stirred for 10 min and TMSOTf (0.10 mL, 0.52 mmol, 4.0 equiv.) was added. The reaction was stirred at 60 °C for 3.5 h and then cooled to room temperature and quenched with saturated aqueous NaHCO_3_ (1 mL). The aqueous layer was extracted with DCM (3 × 5 mL) and the combined organic layers were washed with brine, dried over MgSO_4_, filtered and concentrated under reduced pressure. Purification by flash chromatography on silica gel (DCM/MeOH, 95:5) provided the 1′,2′-*trans* ribo-like nucleoside analogue **26** (63 mg, 77%, dr 20:1) as a white foam. R*_f_* = 0.34 (DCM/MeOH, 95:5); [α]^25^_D_ +35 (*c* 0.2, MeOH); Formula: C_33_H_41_N_3_O_8_Si; MW: 635.79 g/mol; IR (neat) ν_max_ 2957, 2932, 2893, 2858, 1746, 1671 cm^−1^; ^1^H NMR (500 MHz, CD_3_OD) δ 8.17 (d, *J* = 7.6 Hz, 1H), 7.72–7.68 (m, 4H), 7.50–7.41 (m, 6H), 7.11 (d, *J* = 7.5 Hz, 1H), 6.06 (d, *J* = 5.3 Hz, 1H), 5.37 (d, *J* = 5.3 Hz, 1H), 4.27 (t, *J* = 3.9 Hz, 1H), 4.18 (d, *J* = 11.2 Hz, 1H), 4.13 (d, *J* = 11.2 Hz, 1H), 4.09 (dd, *J* = 11.9, 4.0 Hz, 1H), 3.94 (dd, *J* = 11.8, 3.9 Hz, 1H), 2.16 (s, 3H), 2.09 (s, 3H), 2.04 (s, 3H), 1.18 (s, 3H), 1.11 (s, 9H) ppm; NH signals are missing possibly due to exchange in CD_3_OD; ^13^C NMR (126 MHz, CD_3_OD) δ 173.0, 172.2, 171.3, 164.3, 158.1, 145.5, 136.9 (2C), 136.7 (2C), 134.1, 133.5, 131.35, 131.32, 129.12 (2C), 129.09 (2C), 98.2, 90.1, 85.4, 82.5, 67.5, 64.9, 47.2, 27.6 (3C), 24.5, 20.7, 20.6, 20.1, 16.8 ppm; HRMS (ESI) *m/z* [M + H]^+^ calcd for C_33_H_42_N_3_O_8_Si: 636.2736; found 636.2726 (−1.5 ppm).

*(−)-((2S,3R,4R,5R)-4-Acetoxy-5-(6-benzamido-9H-purin-9-yl)-2-(((tert-butyldiphenylsilyl)oxy)methyl)-3-methyltetrahydrofuran-3-yl)methyl acetate* (**27**). To a suspension of *N*^6^-benzoyladenine (44 mg, 0.18 mmol, 2.5 equiv.) in anhydrous DCE (0.9 mL, 0.08 M), bis(trimethylsilyl)acetamide (0.10 mL, 0.40 mmol, 5.5 equiv.) was added at room temperature [33]. The resulting mixture was vigorously stirred for 2 h until a clear solution was obtained. The solution was concentrated under high vacuum and treated with a solution of **25a,b** (40 mg, 74 μmol, 1.0 equiv.) in anhydrous DCE (0.9 mL, 0.08 M). After adding TMSOTf (27 μL, 0.15 mmol, 2.0 equiv.), the resulting solution was heated at reflux for 4 h. The solution was diluted in DCM (5 mL) and washed with a saturated NaHCO_3_ solution (2 mL). The aqueous phase was extracted with DCM (3 × 5 mL) and the combined organic layers were washed with brine, dried over MgSO_4_, filtered and concentrated under reduced pressure. The ^1^H NMR spectrum of the crude reaction mixture indicated a >20:1 dr and only the N^9^-regioisomer. Purification by flash chromatography on silica gel (EtOAc/MeOH, 97:3) provided *N*^6^-benzoyladenine nucleoside analogue **27** (40 mg, 75%) as a white foam. R*_f_* = 0.37 (DCM/MeOH, 95:5); [α]^25^_D_ −15 (*c* 0.8, MeOH); Formula: C_39_H_43_N_5_O_7_Si; MW: 721.89 g/mol; IR (neat) ν_max_ 3268 (br), 3070, 3053, 2956, 2931, 2892, 2857, 1745 cm^−1^; ^1^H NMR (500 MHz, CD_3_OD) δ 8.66 (s, 1H), 8.44 (s, 1H), 8.08 (d, *J* = 7.9 Hz, 2H), 7.69–7.63 (m, 5H), 7.56 (t, *J* = 7.7 Hz, 2H), 7.47–7.38 (m, 4H), 7.31 (t, *J* = 7.5 Hz, 2H), 6.28 (d, *J* = 6.2 Hz, 1H), 6.02 (d, *J* = 6.2 Hz, 1H), 4.34 (t, *J* = 4.4 Hz, 1H), 4.30 (d, *J* = 11.3 Hz, 1H), 4.22 (d, *J* = 11.3 Hz, 1H), 4.08 (dd, *J* = 11.6, 4.3 Hz, 1H), 3.99 (dd, *J* = 11.6, 4.7 Hz, 1H), 2.11 (s, 3H), 2.07 (s, 3H), 1.34 (s, 3H), 1.07 (s, 9H) ppm; NH signals are missing possibly due to exchange in CD_3_OD; ^13^C NMR (126 MHz, CD_3_OD) δ 172.3, 171.5, 168.0, 153.4, 153.3, 151.1, 143.9, 136.73 (2C), 136.65 (2C), 135.0, 134.3, 133.9, 133.8, 131.2, 131.1, 129.8 (2C), 129.4 (2C), 129.0 (2C), 128.9 (2C), 125.0, 88.5, 85.4, 81.4, 67.7, 65.2, 47.4, 27.5 (3C), 20.8, 20.4, 20.1, 17.1 ppm; HRMS (ESI) *m/z* [M + H]^+^ calcd for C_39_H_44_N_5_O_7_Si: 722.3005; found 722.3011 (+0.9 ppm). 

*(+)-((2S,3R,4R,5R)-4-Acetoxy-5-(6-amino-2-chloro-9H-purin-9-yl)-2-(((tert-butyldiphenylsilyl)oxy)methyl)-3-methyltetrahydrofuran-3-yl)methyl acetate* (**28**). To a suspension of 2-chloroadenine (47 mg, 0.28 mmol, 2.5 equiv.) in anhydrous DCE (1.4 mL, 0.08 M), bis(trimethylsilyl)acetamide (0.15 mL, 0.61 mmol, 5.5 equiv.) was added at room temperature. The resulting mixture was vigorously stirred for 2 h until a clear solution was obtained. The solution was concentrated under high vacuum and treated with a solution of **25a,b** (60 mg, 0.11 mmol, 1.0 equiv.) in anhydrous DCE (1.4 mL, 0.08 M). After adding TMSOTf (40 μL, 0.22 mmol, 2.0 equiv.), the resulting solution was heated at reflux for 4 h. The solution was diluted in DCM (5 mL) and washed with a saturated NaHCO_3_ solution (2 mL). The aqueous phase was extracted with DCM (3 × 5 mL) and the combined organic layers were washed with brine, dried over MgSO_4_, filtered and concentrated under reduced pressure. The ^1^H NMR spectrum of the crude reaction mixture indicated a >20:1 dr and only the N^9^-regioisomer. Purification by flash chromatography on silica gel (DCM/MeOH, 95:5) provided 2-chloroadenine nucleoside analogue **28** (61 mg, 85%) as a white foam. R*_f_* = 0.32 (DCM/MeOH, 95:5); [α]^25^_D_ +6.7 (*c* 1.0, MeOH); Formula: C_32_H_38_ClN_5_O_6_Si; MW: 652.22 g/mol; IR (neat) ν_max_ 3319 (br), 3170 (br), 2957, 2932, 2893, 2859, 1744, 1646 cm^−1^; ^1^H NMR (500 MHz, CD_3_OD) δ 8.10 (s, 1H), 7.68–7.63 (m, 4H), 7.46–7.31 (m, 6H), 6.08 (d, *J* = 6.3 Hz, 1H), 5.89 (d, *J* = 6.4 Hz, 1H), 4.30–4.27 (m, 2H), 4.19 (d, *J* = 11.3 Hz, 1H), 4.06 (dd, *J* = 11.6, 4.1 Hz, 1H), 3.94 (dd, *J* = 11.7, 4.3 Hz, 1H), 2.10 (s, 3H), 2.08 (s, 3H), 1.31 (s, 3H), 1.06 (s, 9H) ppm; NH signals are missing possibly due to exchange in CD_3_OD; ^13^C NMR (126 MHz, CD_3_OD) δ 172.4, 171.6, 158.0, 155.4, 151.8, 140.8, 136.8 (2C), 136.7 (2C), 134.2, 133.7, 131.12, 131.11, 129.0 (2C), 128.9 (2C), 119.3, 88.2, 85.4, 81.5, 67.7, 65.2, 47.4, 27.6 (3C), 20.8, 20.5, 20.1, 17.0 ppm; HRMS (ESI) *m/z* [M + H]^+^ calcd for C_32_H_39_ClN_5_O_6_Si^+^: 652.2353; found 652.2344 (–1.3 ppm). 

*(+)-(2R,3R,4R,5S)-2-(4-Acetamido-2-oxopyrimidin-1(2H)-yl)-4-(acetoxymethyl)-5-(hydroxymethyl)-4-methyltetrahydrofuran-3-yl acetate* (**29**). Representative Procedure A: To a solution of 1′,2′-*trans* ribo-like nucleoside **26** (50 mg, 77 μmol, 1.0 equiv.) in anhydrous THF (0.8 mL, 0.1 M), 3HF**‧**NEt_3_ (0.13 mL, 0.79 mmol, 10 equiv.) was added at room temperature and the resulting mixture was stirred for 16 h. Triethylamine (1.10 mL, 7.90 mmol, 100 equiv.) was then added and the mixture was concentrated under reduced pressure. Purification by flash chromatography on silica gel (DCM/MeOH, 95:5) provided the 1′,2′-*trans* ribo-like nucleoside **29** (27 mg, 86%) as a white foam. R*_f_* = 0.30 (DCM/MeOH, 95:5); [α]^25^_D_ +26 (*c* 0.2, MeOH); Formula: C_17_H_2__3_N_3_O_8_; MW: 397.38 g/mol; IR (neat) ν_max_ 3307 (br), 2927, 1742, 1650 cm^−1^; ^1^H NMR (500 MHz, CD_3_OD) δ 8.61 (d, *J* = 7.6 Hz, 1H), 7.45 (d, *J* = 7.5 Hz, 1H), 6.13 (d, *J* = 5.7 Hz, 1H), 5.35 (d, *J* = 5.7 Hz, 1H), 4.22 (t, *J* = 3.6 Hz, 1H), 4.18 (s, 2H), 3.89 (dd, *J* = 12.1, 3.4 Hz, 1H), 3.79 (dd, *J* = 12.0, 3.8 Hz, 1H), 2.18 (s, 3H), 2.09 (s, 3H), 2.07 (s, 3H), 1.19 (s, 3H) ppm; *OH* and NH signals are missing possibly due to exchange in CD_3_OD; ^13^C NMR (126 MHz, CD_3_OD) δ 173.0, 172.4, 171.5, 164.4, 158.3, 146.3, 98.2, 90.0, 85.9, 82.6, 67.7, 62.5, 47.3, 24.5, 20.8, 20.6, 16.4 ppm; HRMS (ESI) *m/z* [M + H]^+^ calcd for C_17_H_2__4_N_3_O_8_: 398.1558; found 398.1552 (–1.6 ppm).

*(−)-((2S,3R,4R,5R)-4-Acetoxy-5-(6-benzamido-9H-purin-9-yl)-2-(hydroxymethyl)-3-methyltetrahydrofuran-3-yl)methyl* (**30**). Following the Representative Procedure A, *N*^6^-benzoyladenine nucleoside analogue **27** (40 mg, 55 μmol, 1.0 equiv.) in anhydrous THF (0.55 mL, 0.10 M), 3HF**·**NEt_3_ (0.14 mL, 0.83 mmol, 15 equiv.) was added and the mixture was stirred for 16 h. Triethylamine (0.77 mL, 5.5 mmol, 100 equiv.) was then added. Purification by flash chromatography on silica gel (DCM/MeOH, 95:5), provided the 1′,2′-*trans* ribo-like *N*^6^-benzoyladenine nucleoside analogue **30** (23 mg, 86%) as a white foam. R*_f_* = 0.54 (DCM/MeOH, 90:10); [α]^25^_D_ −51 (*c* 0.7, MeOH); Formula: C_23_H_2__5_N_5_O_7_; MW: 483.48 g/mol; IR (neat) ν_max_ 3300 (br), 3069, 2927, 2855, 1742 cm^−1^; ^1^H NMR (500 MHz, CD_3_OD) δ 8.82 (s, 1H), 8.70 (s, 1H), 8.09–8.08 (m, 2H), 7.67–7.64 (m, 1H), 7.58–7.55 (m, 2H), 6.35 (d, *J* = 6.7 Hz, 1H), 5.92 (d, *J* = 6.7 Hz, 1H), 4.31–4.25 (m, 3H), 3.93 (dd, *J* = 12.2, 3.1 Hz, 1H), 3.84 (dd, *J* = 12.2, 3.3 Hz, 1H), 2.16 (s, 3H), 2.03 (s, 3H), 1.34 (s, 3H) ppm; OH and NH signals are missing possibly due to exchange in CD_3_OD; ^13^C NMR (126 MHz, CD_3_OD) δ 172.4, 171.5, 168.1, 153.4, 153.1, 151.2, 144.5, 135.0, 133.9, 129.8 (2C), 129.4 (2C), 125.2, 88.4, 86.2, 81.6, 67.9, 63.0, 47.3, 20.8, 20.4, 16.6 ppm; HRMS (ESI) *m/z* [M + H]^+^ calcd for C_23_H_2__6_N_5_O_7_: 484.1827; found 484.1831 (+0.8 ppm).

*(−)-((2S,3R,4R,5R)-4-Acetoxy-5-(6-amino-2-chloro-9H-purin-9-yl)-2-(hydroxymethyl)-3-methyltetrahydrofuran-3-yl)methyl acetate* (**31**). Following the Representative Procedure A, Nucleoside analogue **28** (60 mg, 92 μmol, 1.0 equiv.) in anhydrous THF (0.9 mL, 0.1 M), 3HF**·**NEt_3_ (0.23 mL, 1.4 mmol, 15 equiv.) was added and the mixture was stirred for 16 h. Triethylamine (1.3 mL, 9.2 mmol, 100 equiv.) was then added. Purification by flash chromatography on silica gel (DCM/MeOH, 95:5), provided the 1′,2′-*trans* ribo-like 2-chloroadenine nucleoside analogue **31** (33 mg, 87%) as a white foam. R*_f_* = 0.50 (DCM/MeOH, 90:10); [α]^25^_D_ −75 (*c* 0.2, MeOH); Formula: C_16_H_20_ClN_5_O_6_; MW: 413.82 g/mol; IR (neat) ν_max_ 3326 (br), 2942, 1743, 1618 cm^−1^; ^1^H NMR (500 MHz, CD_3_OD) δ 8.41 (s, 1H), 6.12 (d, *J* = 6.8 Hz, 1H), 5.80 (d, *J* = 6.8 Hz, 1H), 4.27 (d, *J* = 11.2 Hz, 1H), 4.24–4.21 (m, 2H), 3.91 (dd, *J* = 12.4, 3.0 Hz, 1H), 3.80 (dd, *J* = 12.4, 3.2 Hz, 1H), 2.15 (s, 3H), 2.05 (s, 3H), 1.32 (s, 3H) ppm; OH and NH signals are missing possibly due to exchange in CD_3_OD; ^13^C NMR (126 MHz, CD_3_OD) δ 172.5, 171.5, 158.1, 155.2, 151.7, 141.6, 119.4, 88.4, 86.0, 81.4, 67.9, 63.0, 47.2, 20.8, 20.4, 16.6 ppm; HRMS (ESI) *m/z* [M + H]^+^ calcd for C_16_H_21_ClN_5_O_6_^+^: 414.1175; found 414.1169 (−1.5 ppm).

*(+)-4-Amino-1-((2R,3R,4S,5S)-3-hydroxy-4,5-bis(hydroxymethyl)-4-methyltetrahydro furan-2-yl)pyrimidin-2(1H)-one* (**32**). Representative procedure B: To a solution of nucleoside **29** (13 mg, 32 μmol, 1.0 equiv.) in anhydrous MeOH (0.3 mL, 0.1 M), NaOMe (32 μL, 32 μmol, 1.0 equiv., 1.0 M in MeOH) was added at room temperature. The reaction was stirred for 3 h, quenched with amberlite acidic resin (~50 mg) and stirred for 10 min. The mixtrue was filtered with MeOH (~5 mL) and concentrated under reduced pressure. Purification by C18 reverse phase flash chromatography (MeOH/H_2_O) provided 1′,2′-*trans* ribo-like cytosine nucleoside analogue **32** [14] (7 mg, 81%) as a white foam. [α]^25^_D_ +28 (*c* 0.4, CH_3_OH); Formula: C_11_H_17_N_3_O_5_; MW: 271.27 g/mol; IR (neat) ν_max_ 3345, 3217, 2967, 2949, 1649, 1607 cm^−1^; ^1^H NMR (500 MHz, CD_3_OD) δ 8.03 (d, *J* = 7.5 Hz, 1H), 5.91 (d, *J* = 7.5 Hz, 1H), 5.83 (d, *J* = 5.5 Hz, 1H), 4.18 (dd, *J* = 4.5, 3.5 Hz, 1H), 4.11 (d, *J* = 5.5 Hz, 1H), 3.78 (dd, *J* = 11.5, 3.5 Hz, 1H), 3.70 (d, *J* = 11.1 Hz, 1H), 3.67 (dd, *J* = 11.8, 4.7 Hz, 1H), 3.63 (d, *J* = 11.1 Hz, 1H), 1.09 (s, 3H) ppm; *OH* and *NH signals are missing possibly due to exchange in* CD_3_OD; ^13^C NMR (126 MHz, CD_3_OD) δ 167.7, 159.1, 143.1, 95.9, 92.8, 85.5, 83.1, 66.3, 63.1, 48.4, 16.6 ppm; HRMS (ESI) *m/z* [M + Na]^+^ calcd for C_11_H_17_O_5_N_3_Na^+^: 294.1060; found 294.1063 (+0.8 ppm).

*(−)-((2S,3S,4R,5R)-5-(6-Amino-9H-purin-9-yl)-4-hydroxy-3-methyltetrahydrofuran-2,3-diyl)dimethanol* (**33**). Following Representative Procedure B, NaOMe (25 μL, 25 μmol, 1.0 equiv., 1.0 M in MeOH) was added to a solution of nucleoside **30** (12 mg, 25 μmol, 1.0 equiv.) in MeOH (0.25 mL, 0.10 M). Purification by C18 reverse phase flash chromatography (MeOH/H_2_O) provided 1′,2′-*trans* ribo-like adenine nucleoside analogue **33** (6 mg, 82%) as a white foam. R*_f_* = 0.30 (DCM/MeOH, 80:20); [α]^25^_D_ −42 (*c* 0.2, MeOH); Formula: C_12_H_17_N_5_O_4_; MW: 295.30 g/mol; IR (neat) ν_max_ 3332 (br), 3193 (br), 2928, 2882, 1649 cm^−1^; ^1^H NMR (500 MHz, CD_3_OD) δ 8.30 (s, 1H), 8.18 (s, 1H), 6.03 (d, *J* = 7.4 Hz, 1H), 4.66 (d, *J* = 7.5 Hz, 1H), 4.23 (t, *J* = 2.6 Hz, 1H), 3.92 (dd, *J* = 12.6, 2.7 Hz, 1H), 3.80 (d, *J* = 11.0 Hz, 1H), 3.73 (dd, *J* = 12.6, 2.6 Hz, 1H), 3.61 (d, *J* = 11.0 Hz, 1H), 1.26 (s, 3H) ppm; *OH* and *NH signals are missing possibly due to exchange in* CD_3_OD; ^13^C NMR (126 MHz, CD_3_OD) δ 157.6, 153.3, 149.9, 142.4, 121.1, 91.4, 85.9, 80.9, 66.5, 63.9, 48.1, 16.4 ppm; HRMS (ESI) *m/z* [M + H]^+^ calcd for C_12_H_18_N_5_O_4_: 296.1353; found 296.1356 (+1.0 ppm).

*(−)-((2S,3S,4R,5R)-5-(6-amino-2-chloro-9H-purin-9-yl)-4-hydroxy-3-methyltetrahydro furan-2,3-diyl)dimethanol* (**34**). Following Representative Procedure B, NaOMe (24 μL, 24 μmol, 1.0 equiv., 1.0 M in MeOH) was added to a solution of nucleoside **31** (10 mg, 24 μmol, 1.0 equiv.) in MeOH (0.24 mL, 0.10 M). Purification by C18 reverse phase flash chromatography (MeOH/H_2_O) provided 1′,2′-*trans* ribo-like 2-chloroadenine nucleoside analogue **34** (6.4 mg, 80%) as a white foam. R*_f_* = 0.50 (DCM/MeOH, 80:20); [α]^25^_D_ −25 (*c* 0.2, MeOH); Formula: C_12_H_16_ClN_5_O_4_; MW: 329.74 g/mol; IR (neat) ν_max_ 3322 (br), 3186 (br), 2940, 2884, 1653 cm^−1^; ^1^H NMR (500 MHz, CD_3_OD) δ 8.30 (s, 1H), 5.98 (d, *J* = 7.3 Hz, 1H), 4.61 (d, *J* = 7.3 Hz, 1H), 4.22 (t, *J* = 2.9 Hz, 1H), 3.90 (dd, *J* = 12.5, 2.9 Hz, 1H), 3.79 (d, *J* = 11.0 Hz, 1H), 3.74 (dd, *J* = 12.5, 3.0 Hz, 1H), 3.60 (d, *J* = 11.0 Hz, 1H), 1.24 (s, 3H) ppm; *OH* and *NH signals are missing possibly due to exchange in* CD_3_OD; ^13^C NMR (126 MHz, CD_3_OD) δ 158.2, 155.0, 151.4, 142.4, 119.9, 91.1, 85.8, 81.1, 66.4, 63.8, 48.1, 16.4 ppm; HRMS (ESI) *m/z* [M + H]^+^ calcd for C_12_H_17_ClN_5_O_4_: 330.0964; found 330.0958 (−1.7 ppm).

*(−)-(3R,4S,5R)-4-((tert-Butyldimethylsilyl)oxy)-5-(((tert-butyldimethylsilyl)oxy)methyl)-3-methyl-3-vinyldihydrofuran-2(3H)-one* (**S2**). To a solution of secondary alcohol **40** (4.00 g, 14.0 mmol, 1.00 equiv.) in anhydrous DCM (42 mL, 0.33 M), 2,6 Lutidine (4.04 mL, 34.9 mmol, 2.50 equiv.) and TBSOTf (4.81 mL, 2.09 mmol, 1.50 equiv.) were added at 0 °C. The resulting mixture was gradually warmed to room temperature and stirring was continued for overnight. A saturated aqueous solution of NaHCO_3_ (~20 mL) was added and the mixture was extracted with CH_2_Cl_2_ (3 × 40 mL). The combined organic layers were washed with brine, dried over MgSO_4_, filtered and condensed under reduced pressure. The residue was purified by flash chromatography on silica gel (Hexanes/EtOAc, 90:10) to provide bis-silylated ethers **S2** (4.2 g, 75%) as a colorless oil. R*_f_*= 0.28 (Hexanes/EtOAc, 9:1); [α]^25^_D_ −60 (*c* 1.2, CH_2_Cl_2_); IR (neat) ν_max_ 2954, 2930, 2858, 1786 cm^−1^; Formula: C_20_H_40_O_4_Si_2_; MW: 400.70 g/mol; ^1^H NMR (500 MHz, CDCl_3_): δ 5.94 (dd, *J* = 17.7, 10.7 Hz, 1H), 5.24 (dd, *J* = 10.7, 0.9 Hz, 1H), 5.18 (d, *J* = 17.6 Hz, 1H), 4.27 (d, *J* = 8.1 Hz, 1H), 4.28–3.99 (m, 1H), 3.98 (dd, *J* = 12.3, 1.8 Hz, 1H), 3.74 (dd, *J* = 12.3, 2.5 Hz, 1H), 1.33 (s, 3H), 0.90 (s, 9H), 0.88 (s, 9H), 0.11 (s, 3H), 0.10 (s, 3H), 0.07 (s, 3H), 0.06 (s, 3H) ppm. ^13^C NMR (126 MHz, CDCl_3_) δ 176.6, 134.1, 116.8, 82.1, 75.0, 59.6, 51.6, 25.9 (3C), 25.8 (3C), 21.3, 18.4, 18.1, −4.2, −4.7, −5.2, −5.4 ppm; HRMS (ESI) *m/z* [M + H]^+^ calcd for C_20_H_41_O_4_Si_2_: 401.2543; found 401.2539 (+0.2 ppm) and [M+NH_4_]^+^ calcd for C_20_H_44_NO_4_Si_2_: 418.2809; found 418.2800 (−0.7 ppm).

*(+)-(3S,4S,5R)-4-((tert-Butyldimethylsilyl)oxy)-5-(((tert-butyldimethylsilyl)oxy)methyl)-3-methyl-2-oxotetrahydrofuran-3-carbaldehyde* (**41**). The alkene **S2** (761 mg, 1.90 mmol, 1.00 equiv.) was dissolved in CH_2_Cl_2_ (114 mL, 0.02 M) and the mixture was cooled to −78 °C. Ozone was bubbled through the reaction mixture until a blue color appeared at which point the ozone inlet was changed for a N_2_ inlet and bubbling was continued for 15 min. The solution was then purged with nitrogen to remove excess of ozone. Triethylamine (0.265 mL, 1.90 mmol, 1.00 equiv.) was added and the mixture was stirred for 30 min at −78 °C. Then, it was gradually warmed to room temperature and stirring was continued for 1 h. The reaction mixture was filtered over MgSO_4_ and condensed under reduced pressure. The crude was purified by flash chromatography on silica gel (Hexanes/EtOAc, 100:0 to 70:30) to provide the aldehyde **41** (663 mg, 87%) as colourless oil. R*_f_* = 0.67 (EtOAc/Hexanes, 1:4); [α]^25^_D_: +37 (*c* 5.4, in CH_2_Cl_2_); IR: (neat) ν_max_ 2954, 2930, 2886, 2858, 1789, 1729, 1472, 1463, 1254 cm^−1^. Formula: C_19_H_38_O_5_Si_2_; MW: 402.67 g.mol^−1^; NMR ^1^H: (500 MHz, CDCl_3_) δ 9.57 (s, 1H), 4.54 (d, *J* = 6.8 Hz, 1H), 4.35 (appdt, *J* = 6.8, 2.2 Hz, 1H), 4.00 (dd, *J* = 12.3, 2.1 Hz, 1H), 3.77 (dd, *J* = 12.3, 2.3 Hz, 1H), 1.49 (s, 3H), 0.88 (s, 9H), 0.86 (s, 9H), 0.11 (s, 3H), 0.09 (s, 3H), 0.08 (s, 3H), 0.06 (s, 3H) ppm; NMR ^13^C: (126 MHz, CDCl_3_) δ 195.9, 172.7, 83.7, 76.8, 60.5, 59.9, 25.9 (3C), 25.6 (3C), 18.4, 17.9, 15.4, −4.5, −4.8, −5.2, −5.4 ppm; HRMS (ESI): *m/z* [M + H]^+^ calcd for C_19_H_39_O_5_Si_2_: 403.2331, found: 403.2329 (−0.50 ppm). 

*(3R,4S,5R)-4-((tert-Butyldimethylsilyl)oxy)-5-(((tert-butyldimethylsilyl)oxy)methyl)-3-(hydroxymethyl)-3-methyltetrahydrofuran-2-ol* (**42a**,**b**). To a solution of aldehyde **41** (2.30 g, 5.71 mmol, 1.00 equiv.) in anhydrous THF (57 mL, 0.10 M), Red-Al (3.57 mL, 11.4 mmol, 2.00 equiv.) was added at −40 °C and stirring was continued for 40 min. at −40 °C. The reaction was then cooled to −78 °C and was quenched by the addition of few drops (~0.5 mL) of saturated Rochelle salt solution. The stirring was continued for 10 min at −78 °C and it was then gradually warmed to room temperature. THF (~20 mL) and saturated Rochelle salt solution (~15 mL) were added. The resulting biphasic mixture was stirred vigorously for an hour. The layers were separated, the aqueous layer was extracted with Et_2_O (3 × 40 mL), and the combined organic layers were washed with brine, dried over MgSO_4_, filtered and concentrated under reduced pressure. The residue was purified by flash chromatography on silica gel (Hexanes/EtOAc, 4:1) to give the lactol **42a**,**b** (1.6 g, 69%, dr 1:2) as a colorless oil. R*_f_* = 0.24 (Hexanes/EtOAc, 4:1); IR (neat) ν_max_ 3329, 3200, 2930, 1645, 1600 cm^−1^; Formula: C_19_H_42_O_5_Si_2_; MW: 406.70 g/mol; ^1^H NMR (500 MHz, CDCl_3_): δ 5.36 (dd, *J* = 8.9, 2.3 Hz, 1H, minor (OH, D_2_O exchange), 5.04 (d, *J* = 8.6 Hz, 1H, minor), 4.92 (d, *J* = 8.8 Hz, 1H, major), 4.25 (d, *J* = 5.5 Hz, 1H, major), 4.08 (d, *J* = 6.3 Hz, 1H, minor), 4.02–3.95 (m, 2H, major and minor), 3.89 (d, *J* = 11.8 Hz, 1H, minor), 3.83 (dd, *J* = 11.6, 2.8 Hz, 1H, minor), 3.80–3.74 (m, 2H, major and minor), 3.74–3.67 (m, 2H, minor and major), 3.65 (dd, *J* = 11.0, 2.3 Hz, 1H, major), 3.59–3.52 (m, 2H, major and OH (D_2_O exchange)), 1.11 (s, 3H, major), 1.01 (s, 3H, minor), 0.93 (s, 9H, major), 0.90 (s, 9H, minor), 0.89 (s, 18H, major and minor), 0.12 (s, 6H, major), 0.11 (s, 12H, minor), 0.07 (s, 3H, major), 0.06 (s, 3H, major) ppm; OH signals are missing possibly due to exchange in CDCl_3_ ppm. ^13^C NMR (126 MHz, CDCl_3_) δ 105.7 (minor), 102.2 (major), 85.4 (major), 84.6 (minor), 78.9 (minor), 78.7 (major), 66.7 (major), 65.8 (minor), 62.6 (minor), 61.6 (major), 51.4 (major), 49.0 (minor), 26.00 (3C, minor), 25.96 (3C, major), 25.84 (3C, minor), 25.80 (3C, major), 21.2 (minor), 18.43 (minor), 18.40 (major), 18.0 (minor), 17.9 (major), 16.0 (major), −4.3 (major), −4.39 (minor), −4.43 (major), −4.6 (major), −5.1 (minor), −5.39 (major), −5.40 (2C, minor) ppm; HRMS (ESI) *m/z* [M+NH_4_]^+^ calcd for C_19_H_46_NO_5_Si_2_: 424.2915; found 424.2904 (−1.6 ppm).

*((3R,4S,5R)-2-(Benzoyloxy)-4-((tert-butyldimethylsilyl)oxy)-5-(((tert-butyldimethylsilyl)oxy)methyl)-3-methyltetrahydrofuran-3-yl)methyl benzoate* (**43a,b**). To a solution of lactol **42a,b** (3.30 g, 8.11 mmol, 1.00 equiv.) in CH_2_Cl_2_ (45 mL, 0.18 M), pyridine (3.94 mL, 48.7 mmol, 6.00 equiv.) and DMAP (99 mg, 0.81 mmol, 0.10 equiv.) were added. After cooling the resulting mixture to 0 °C, BzCl (4.71 mL, 40.6 mmol, 5.00 equiv.) was added dropwise. The reaction mixture was gradually warmed to room temperature and stirred for 16 h. The mixture was cooled to 0 °C and ethylenediamine (1.36 mL, 20.3 mmol, 2.50 equiv.) was added and stirring was continued for 1 h at 0 °C. The reaction mixture was then diluted with hexanes (40 mL) and passed through a celite using Et_2_O. The filtrates were concentrated under reduced pressure. The residue was purified by flash chromatography on silica gel (Hexanes/EtOAc, 4:1) to afford the **43a,b** (3.8 g, 76%, dr 1.2:1) as a mixture of anomers as a white foam. R*_f_*= 0.34 (Hexanes/EtOAc, 4:1); IR (neat) ν_max_ 2954, 2929, 2885, 2857, 1722, 1261 cm^−1^; Formula: C_33_H_50_O_7_Si_2_; MW: 614.92 g/mol; ^1^H NMR (500 MHz, CDCl_3_): δ 8.11–8.03 (m, 4H, major), 8.00–7.96 (m, 2H, minor), 7.62–7.50 (m, 5H, minor), 7.49–7.36 (m, 9H, major and minor), 6.54 (s, 1H, major), 6.37 (s, 1H, minor), 4.61 (d, *J* = 11.1 Hz, 1H, minor), 4.56 (d, *J* = 11.1 Hz, 1H, minor), 4.58–4.54 (m, 2H, major and minor), 4.40 (d, *J* = 7.4 Hz, 1H, major), 4.27–4.23 (m, 1H, minor), 4.19 (d, *J* = 3.1 Hz, 1H, minor), 4.09–4.03 (m, 1H, major), 3.91 (dd, *J* = 11.6, 2.4 Hz, 1H, major), 3.88–3.84 (m, 2H, major and minor), 3.72 (dd, J = 11.6, 4.4 Hz, 1H, major), 1.32 (s, 6H, major and minor), 0.97 (s, 9H, minor), 0.94 (s, 9H, major), 0.93 (s, 9H, minor), 0.79 (s, 9H, major), 0.19 (s, 3H, major), 0.14 (s, 6H, major and minor), 0.13 (s, 3H, minor), 0.11 (s, 6H, major and minor), −0.01 (s, 3H, major), −0.08 (s, 3H, minor) ppm. ^13^C NMR (126 MHz, CDCl_3_) δ 166.6 (minor), 166.5 (major), 165.6 (major), 165.5 (minor), 133.34 (major), 133.31 (minor), 133.2 (major), 133.0 (minor), 130.3 (minor), 130.13 (major), 130.06 (minor), 130.0 (3C, 2minor and 1major), 129.9 (2C, major), 129.8 (2C, major), 129.61 (2C, minor), 128.6 (2C, major), 128.5 (4C, major and minor), 128.4 (2C, minor), 102.4 (minor), 100.5 (major), 89.9 (minor), 85.5 (major), 77.8 (minor), 76.7 (major), 66.3 (major), 65.4 (minor), 63.1 (major), 62.6 (minor), 50.1 (minor), 49.8 (major), 26.1 (3C, minor), 26.0 (3C, major), 25.9 (3C, major), 25.8 (3C, minor), 20.8 (minor), 18.6 (minor), 18.5 (major), 18.08 (major), 18.07 (minor), 16.9 (major), −4.1 (major), −4.2 (minor), −4.3 (major), −4.8 (major), −5.2 (minor), −5.28 (minor), −5.34 (2C major and minor) ppm HRMS (ESI) *m/z* [M-OBz]^+^ calcd for C_26_H_45_O_5_Si_2_: 493.2806; found 493.2805; and [M+NH_4_]^+^ calcd for C_33_H_54_NO_7_Si_2_: 632.3439; found 632.3439 (−0.2 ppm).

*(−)((2R,3R,4S,5R)-4-((tert-Butyldimethylsilyl)oxy)-5-(((tert-butyldimethylsilyl)oxy)methyl)-2-(2,6-dichloro-9H-purin-9-yl)-3-methyltetrahydrofuran-3-yl)methyl benzoate* (**44**). To a solution of the benzoylated furanosides **43a**,**b** (434 mg, 0.706 mmol, 1.00 equiv.) in anhydrous MeCN (2.8 mL, 0.25 M), 2,6-dichloropurine (147 mg, 0.776 mmol, 1.10 equiv.) was added at room temperature. The resulting mixture was cooled to −10 °C and DBU (0.316 mL, 2.12 mmol, 3.00 equiv.) was added followed by dropwise addition of TMSOTf (0.520 mL, 2.82 mmol, 4.00 equiv.). The stirring was continued at −10 °C for 3 h. The mixture was warmed to room temperature, and a saturated solution of NaHCO_3_ (~8 mL) was added. The aqueous layer was extracted with CH_2_Cl_2_ (3 × 20 mL) and the combined organic layers were washed with brine, dried over MgSO_4_, filtered and concentrated under reduced pressure. The crude was purified by flash chromatography on silica gel (CH_2_Cl_2_/MeOH, 100:0 to 70:30) to provide the pure product **44** (375 mg, 78%, β:α = 10:1) as brown gum. R*_f_* = 0.87 (MeOH/CH_2_Cl_2_, 1:9); [α]^25^_D_: −10 (*c* 4.9, MeOH); IR: (neat) ν_max_ 2955, 2930, 2898, 2858, 1724, 1590, 1553, 1470, 1464, 1452, 1356, 1255, 1214 cm^−1^; Formula: C_31_H_46_Cl_2_N_4_O_5_Si_2_; MW: 680.80 g.mol^−1^; NMR ^1^H: (500 MHz, CDCl_3_) δ 8.90 (s, 1H), 8.20 (s, 2H), 7.61 (t, *J* = 7.4 Hz, 1H), 7.52 (t, *J* = 7.7 Hz, 2H), 6.61 (s, 1H), 4.60 (d, *J* = 11.7 Hz, 1H), 4.50 (d, *J* = 7.2 Hz, 1H), 4.48 (d, *J* = 11.6 Hz, 1H), 4.15 (d, *J* = 12.1 Hz, 1H), 4.07 (appd, *J* = 8.4 Hz, 1H), 3.90 (d, *J* = 12.1 Hz, 1H), 1.01 (s, 9H), 0.93 (s, 9H), 0.87 (s, 3H), 0.21 (s, 3H), 0.21 (s, 3H), 0.14 (s, 3H), 0.12 (s, 3H) ppm; NMR ^13^C: (126 MHz, CDCl_3_) δ 166.7, 153.1, 152.8, 151.9, 145.0, 133.5, 131.0, 129.9 (2C), 129.7, 128.8 (2C), 88.2, 84.3, 74.6, 66.5, 61.0, 50.1, 26.4 (3C), 25.9 (3C), 18.8, 18.1, 17.5, −4.1, −4.3, −5.0, −5.1 ppm; HRMS (ESI) *m/z*: [M + H]^+^ calcd for C_31_H_47_Cl_2_N_4_O_5_Si_2_: 681.2457, found: 681.2450 (−1.0 ppm).

*(+)((2R,3R,4S,5R)-2-(6-Amino-2-chloro-9H-purin-9-yl)-4-((tert-butyldimethylsilyl)oxy)-5-(((tert-butyldimethylsilyl)oxy)methyl)-3-methyltetrahydrofuran-3-yl)methanol* (**S3**). To a stirred solution of the nucleoside **44** (199 mg, 0.292 mmol, 1.00 equiv.) in anhydrous MeOH (1.2 mL, 0.20 M) in a high-pressure flask, NH_3_(g) was bubbled until saturation at room temperature. The reaction mixture was then stirred at 80 °C for 24 h. The mixture was then diluted with MeOH and concentrated under reduced pressure. The crude was purified by flash chromatography on silica gel (CH_2_Cl_2_/MeOH, 100/0 to 94/6) to provide the nucleoside analogue **S3** (141 mg, 87%) as white foam. R*_f_*
**=** 0.66 (EtOAc/Hexanes, 1:4); [α]^25^_D_: + 12 (*c* 0.8, in MeOH); IR: (neat) ν_max_ 3324, 3189, 2955, 2930, 2896, 2858, 1642, 1594, 1463, 1346, 1316, 1253, 1215 cm^−1^_;_ Formula: C_24_H_44_ClN_5_O_4_Si_2_; MW: 558.26 g.mol^−1^; NMR ^1^H: (500 MHz, CDCl_3_) δ 8.40 (s, 1H), 6.30 (s, 1H), 4.37 (d, *J* = 7.6 Hz, 1H), 4.06 (dd, *J* = 11.8, 2.0 Hz, 1H), 4.01 (dt, *J* = 7.6, 2.3 Hz, 1H), 3.89 (s, 2H), 3.86 (dd, *J* = 11.8, 2.5 Hz, 1H), 0.97 (s, 9H), 0.93 (s, 9H), 0.71 (s, 3H), 0.17 (s, 3H), 0.16 (s, 3H), 0.14 (s, 3H), 0.13 (s, 3H) ppm; *OH* and *NH signals are missing possibly due to exchange in* CDCl_3;_ NMR ^13^C: (126 MHz, CDCl_3_) δ 155.6, 150.3, 145.8, 139.3, 114.2, 89.3, 85.0, 77.5, 66.2, 61.1, 50.6, 26.3 (3C), 25.9 (3C), 18.7, 18.0, 17.3, −4.1, −4.4, −5.1, −5.1 ppm; HRMS (ESI)**:**
*m/z* [M + H]^+^ calcd for C_24_H_45_ClN_5_O_4_Si_2_: 558.2693, found: 558.2686 (−1.25 ppm).

*(−)-((2R,3S,4R,5R)-5-(6-Amino-2-chloro-9H-purin-9-yl)-3-hydroxy-4-methyltetra hydrofuran-2,4-diyl)dimethanol* (**45**). To a solution of nucleoside **S3** (68 mg, 0.12 mmol, 1.0 equiv) in THF (1.2 mL, 0.1 M) was added HF⸱Et_3_N (0.30 mL, 1.8 mmol, 15 equiv.) at room temperature. The reaction mixture was stirred for 18 h. It was then cooled to 0 °C and triethylamine (1.70 mL, 12.2 mmol, 100 equiv.) was added. The mixture was concentrated under reduced pressure. The crude was purified by reverse phase C18 flash chromatography (MeOH/H_2_O, 0:100 to 4:6) to provide nucleoside analogue **45** (30 mg, 75%) as white foam. R*_f_* = 0.17 (CH_2_Cl_2_/CH_3_OH, 90:10); [α]^25^_D_ –10 (*c* 1.0, CH_3_OH); Formula: C_12_H_16_ClN_5_O; MW: 329.74 g/mol; IR (neat) νmax 3339, 3199, 2936, 1653, 1594 cm^−1^; 1H NMR (500 MHz, CD_3_OD) δ 8.48 (s, 1H), 6.27 (s, 1H), 4.36 (d, *J* = 8.8 Hz, 1H), 4.07 (ddd, *J* = 8.8, 3.6, 2.2 Hz, 1H), 3.97 (dd, *J* = 12.4, 2.2 Hz, 1H), 3.89–3.84 (m, 2H), 3.77 (d, *J* = 11.3 Hz, 1H), 0.70 (s, 3H) ppm; OH and NH signals are missing possibly due to exchange in CD_3_OD; ^13^C NMR (125 MHz, CD_3_OD) δ 158.1, 155.3, 151.5, 140.9, 119.0, 90.6, 85.6, 76.3, 65.4, 61.7, 51.7, 17.3 ppm; HRMS (ESI): *m/z* [M + Na]^+^ calcd for: C_12_H_16_ClN_5_NaO_4_: 352.0783; found 352.0783 (−0.096 ppm).

*(−)-((2R,3S,4R,5R)-5-(6-Amino-9H-purin-9-yl)-3-hydroxy-4-methyltetrahydrofuran-2,4-diyl)dimethanol* (**46**). To a solution of nucleoside **45** (70 mg, 0.21 mmol, 1.0 equiv) in methanol (10 mL, 0.021 M), palladium (10 wt.%) on activated carbon (90 mg, 85 μmol, 0.40 equiv.) was added. The reaction mixture was degassed and flushed using a hydrogen-filled balloon. The resulting reaction was stirred for overnight at 40 °C. The reaction mixture was filtered through Celite^®^, washed with methanol, and filtrates were concentrated under reduced pressure to provide **46** (49 mg, 78%). The crude was pure by NMR and used for the next step without purification. R*_f_* = 0.27 (CH_2_Cl_2_/CH_3_OH, 4:1); [α]^25^_D_ −60 (c 0.4, CH_3_OH); IR (neat) ν_max_ 3427, 2954, 2929, 2886, 2857, 1472, 1252 cm^−1^; Formula: C_12_H_17_N_5_O_4_; MW: 295.29 g/mol; ^1^H NMR (500 MHz, CD_3_OD): δ 8.51 (s, 1H), 8.18 (s, 1H), 6.35 (s, 1H), 4.37 (d, *J* = 8.7 Hz, 1H), 4.08 (ddd, *J* = 8.7, 3.1, 2.3 Hz, 1H), 3.99 (dd, *J* = 12.4, 2.3 Hz, 1H), 3.89–3.82 (m, 2H), 3.77 (d, *J* = 11.2 Hz, 1H), 0.67 (s, 3H) OH and NH signals are missing possibly due to exchange in CD_3_OD ppm; ^13^C NMR (126 MHz, CD_3_OD) δ 157.4, 153.7, 150.3, 141.6, 120.1, 90.7, 85.6, 76.1, 65.5, 61.5, 51.7, 17.3 ppm.; HRMS (ESI) *m/z* [M + H]^+^ calcd for C_12_H_18_N_5_O_4_: 296.1359; found 296.1348 (−1.8 ppm).

General Procedure C: Prior to the reaction, nucleoside analogue (**29**–**31** and **45**–**47**), salicyl phosphorochloridite (SalPCl) and tributylammonium pyrophosphate [(Bu_3_HN)_2_H_2_P_2_O_7_] were respectively dried under reduced pressure in 10 mL, 5 mL, 5 mL flasks for 1 h. To a solution of (Bu_3_HN)_2_H_2_P_2_O_7_ (1.2–2.5 equiv.) in anhydrous DMF (0.1 M), NBu_3_ (0.25 M) was added under nitrogen atmosphere and the mixture was stirred until (5 min) it became homogenious. The reaction mixture then was injected into a 5 mL flask containing SalPCl (1.2–2.5 equiv.) and the resulting mixture was stirred at room temperature for 30 min. The mixture was then transferred to a flask containing nucleoside **29**–**31** and **45**–**47** (1.0 equiv.) and the resulting mixture was stirred for 1.5 h. A solution of iodine (3% in Pyr: H_2_O 9:1 wV) was injected dropwise into the solution until a permanent brown color was persisted (~0.5 mL) and the resulting mixture was stirred for 15 min. Water (1.5 mL) was added and the solution was stirred for 1.5 h to provide the desired C5′-triphosphate which was detected by TLC (*i-*PrOH: NH_4_OH: H_2_O, 5:3:2). The reaction mixture was transferred into a centrifuge tube using 15 mL of EtOH. A solution of 3M NaCl was added dropwise until the reaction mixture became cloudy (~0.5 mL) and was cooled to –78 °C for 1 h. Centrifugation was conducted at 10 °C with 3200 rpm for 20 min and the resulting liquid phase was then transferred to a 50 mL Erlenmeyer flask. The resulting solid (residue) inside the centrifuge tube was air dried for 15 min. The residue was purified by reverse phase C18 flash chromatography (MeCN in 20 mM triethylammonium acetate (TEAAc) buffer, pH = 7) to provide corresponding nucleoside triphosphate triethylammonium salt, which was then lyophilized to provide pure solid nucleoside triphosphate as a white powder.

General Procedure D: If the nucleside analogues hydroxyl or amine functional groups are protected with acetyl (Ac) or benzoyl (Bz) follow the Gerenaral Procedure C until centrifiguation. Residue (solid) inside the centrifuge tube was dissolved in NH_4_OH (0.02 M) and was stirred for overnight. The mixture was concentrated under reduced pressure. The residue was purified by reverse phase C18 flash chromatography (MeCN in 20 mM triethylammonium acetate buffer, pH = 7) to provide corresponding nucleoside triphosphate triethylammonium salt, which was then lyophilized to provide pure solid nucleoside triphosphate as a white powder. 

*((2S,3S,4R,5R)-5-(4-Amino-2-oxopyrimidin-1(2H)-yl)-4-hydroxy-3-(hydroxymethyl)-3-methyltetrahydrofuran-2-yl)methyl tetrahydrogen triphosphate* **2** (LCB-2330). Following general procedure D, (Bu_3_HN)_2_H_2_P_2_O_7_ (0.11 g, 0.20 mmol, 2.5 equiv.), NBu_3_ (0.28 mL, 0.25 M), SalPCl (40 mg, 0.20 mmol, 2.5 equiv.) and nucleoside analogue **29** (36.0 mg, 0.079 mmol, 1.00 equiv.) in anhydrous DMF (0.8 mL, 0.1 M) were employed in the phosphorylation. The final mixture was concentrated under reduced pressure. Purification by reverse phase C18 flash chromatography (flow rate of 10 mL/min., gradient run of acetonitrile from 0 to 10% in 20 mM TEAAc, pH = 7) provided nucleoside triphosphate triethylammonium salt **2** (LCB-2330) (20 mg, 28%) as a white powder. Formula: C_11_H_20_N_3_O_14_P_3_; MW: 511.21 g/mol; ^1^H NMR (500 MHz, D_2_O, signals for triethylammonium denoted by *) δ 8.38 (d, *J* = 8.0 Hz, 1H), 6.42 (d, *J* = 8.0 Hz, 1H), 6.11 (d, *J* = 6.5 Hz, 1H), 4.42–4.41 (m, 1H), 4.32 (d, *J* = 6.8 Hz, 1H), 4.30–4.27 (m, 1H), 4.16–4.13 (m, 1H), 3.78 (d, *J* = 11.6 Hz, 1H), 3.64 (d, *J* = 11.5 Hz, 1H), 3.22 (q, *J* = 7.3 Hz, 16H*), 1.29 (t, *J* = 7.3 Hz, 24H*), 1.17 (s, 3H) ppm; *OH* and *NH_2_ signals are missing possibly due to exchange in* D_2_O; ^13^C NMR (126 MHz, D_2_O, signals for triethylammonium denoted by *) δ 164.4, 155.9, 142.3, 88.9, 82.5 (d, *J* = 9.3 Hz), 80.4, 66.2, 64.5, 46.9, 46.6*, 42.2, 15.2, 8.2* ppm; ^31^P NMR (162 MHz, D_2_O) δ −10.69 (br s, 1P), −11.94 (s, 1P), −23.14 (br s, 1P) ppm; HRMS (ESI) *m/z* [M − H]^−^ calcd for C_11_H_19_N_3_O_14_P_3_: 510.0085; found 510.0087 (+0.4 ppm).

*((2S,3S,4R,5R)-5-(6-Amino-9H-purin-9-yl)-4-hydroxy-3-(hydroxymethyl)-3-methyltetrahydrofuran-2-yl)methyl tetrahydrogen triphosphate* **3** (LCB-2332). Following general procedure D, (Bu_3_HN)_2_H_2_P_2_O_7_ (0.10 g, 0.18 mmol, 2.5 equiv.), NBu_3_ (0.29 mL, 0.25 M), SalPCl (37 mg, 0.18 mmol, 2.5 equiv.) and nucleoside analogue **30** (35.0 mg, 0.072 mmol, 1.00 equiv.) in anhydrous DMF (0.6 mL, 0.1 M) were employed in the phosphorylation. The final mixture was concentrated under reduced pressure. Purification by reverse phase C18 flash chromatography (flow rate of 10 mL/min., gradient run of acetonitrile from 0 to 10% in 20 mM TEAAc, pH =7) provided nucleoside triphosphate **3** (LCB-2332) triethylammonium salt (18 mg, 27%) as a white powder. Formula: C_12_H_20_N_5_O_13_P_3_; MW: 535.24 g/mol; ^1^H NMR (500 MHz, D_2_O, signals for triethylammonium denoted by *) δ 8.66 (br s, 1H), 8.25 (s, 1H), 6.20 (d, *J* = 7.0 Hz, 1H), 4.76–4.73 (m, 1H), 4.46 (s, 1H), 4.25–4.14 (m, 1H), 3.85 (d, *J* = 11.5 Hz, 1H), 3.73 (d, *J* = 11.5 Hz, 1H), 3.19 (q, *J* = 7.3 Hz, 18H*), 1.27 (t, *J* = 7.3 Hz, 27H*), 1.24 (s, 3H) ppm; *OH* and *NH_2_ signals are missing possibly due to exchange in* D_2_O; ^13^C NMR (126 MHz, D_2_O, signals for triethylammonium denoted by *) δ 154.7, 151.6, 149.4, 140.5, 118.8, 86.9, 82.8 (d, *J* = 10.2 Hz), 80.1, 66.3 (d, *J* = 6.4 Hz), 64.5, 47.0, 46.5*, 15.1, 8.2* ppm; ^31^P NMR (162 MHz, D_2_O) δ –11.68 (br s, 2P), –23.60 (br s, 1P) ppm; HRMS (ESI) *m/z* [M − H]^−^ calcd for C_12_H_19_N_5_O_13_P_3_: 534.0198; found 534.0203 (+1.0 ppm).

*((2S,3S,4R,5R)-5-(6-Amino-2-chloro-9H-purin-9-yl)-4-hydroxy-3-(hydroxymethyl)-3-methyltetrahydrofuran-2-yl)methyl tetrahydrogen triphosphate* **4** (LCB-2337). Following general procedure D, (Bu_3_HN)_2_H_2_P_2_O_7_ (73.0 mg, 0.133 mmol, 2.20 equiv.), NBu_3_ (0.35 mL, 0.25 M), SalPCl (27.0 mg, 0.133 mmol, 2.20 equiv.) and nucleoside analogue **31** (25.0 mg, 0.06 mmol, 1.00 equiv.) in anhydrous DMF (0.5 mL, 0.1 M) were employed in the phosphorylation. The final mixture was concentrated under reduced pressure. Purification by reverse phase C18 flash chromatography (flow rate of 10 mL/min., gradient run of acetonitrile from 0 to 10% in 20 mM TEAAc, pH = 7) provided nucleoside triphosphate **4** (LCB-2337) triethylammonium salt (30 mg, 51%) as a white powder. Formula: C_12_H_19_ClN_5_O_13_P_3_; MW: 569.68 g/mol; ^1^H NMR (500 MHz, D_2_O, signals for triethylammonium denoted by *) δ 8.60 (s, 1H), 6.12 (d, *J* = 7.1 Hz, 1H), 4.75–4.70 (m, 1H), 4.46 (q, *J* = 3.3 Hz, 1H), 4.26–4.12 (m, 2H), 3.84 (d, *J* = 11.5 Hz, 1H), 3.72 (d, *J* = 11.5 Hz, 1H), 3.20 (q, *J* = 7.3 Hz, 22H*), 1.28 (t, *J* = 7.3 Hz, 31H*), 1.24 (s, 3H) ppm; OH and NH_2_ signals are missing possibly due to exchange in D_2_O; ^13^C NMR (126 MHz, D_2_O, signals for triethylammonium denoted by *) δ 156.3, 153.8, 150.7 (Brs), 140.2, 117.54 (Brs), 86.9, 82.9 (d, *J* = 10.1 Hz), 80.2, 66.23 (d, *J* = 6.2 Hz), 64.5, 47.1, 46.6*, 15.2, 8.2* ppm (J values result from ^13^C−31P coupling and were assigned when possible); ^31^P NMR (162 MHz, D_2_O) δ −6.34 (d, *J* = 18.7 Hz, 1P), −11.71 (d, *J* = 19.4 Hz, 1P), −22.58 (t, *J* = 19.0 Hz, 1P) ppm; HRMS (ESI) *m/z* [M − H]⁻ calcd for C_12_H_18_ClN_5_O_13_P_3_: 567.9808; found 567.9808 (−0.09 ppm). 

*((2R,3S,4R,5R)-5-(4-Amino-2-oxopyrimidin-1(2H)-yl)-3-hydroxy-4-(hydroxymethyl)-4-methyltetrahydrofuran-2-yl)methyl triphosphate:* **5** (LCB-2289). Following general procedure C, (Bu_3_HN)_2_H_2_P_2_O_7_ (162 mg, 295 μmol, 2.00 equiv.), NBu_3_ (0.6 mL, 0.25 M), SalPCl (60.0 mg, 295 μmol, 2.00 equiv.) and nucleoside analogue **47** [13] (40.0 mg, 147 μmol, 1.00 equiv.) in anhydrous DMF (1 mL, 0.1 M) were employed in the phosphorylation. The final mixture was concentrated under reduced pressure. Purification by reverse phase C18 flash chromatography (flow rate of 8 mL/min., gradient run of acetonitrile from 0 to 6% in 20 mM TEAAc, pH = 7) provided nucleoside triphosphate **5** (LCB-2289) triethylammonium salt (7 mg, 5%) as a white powder. Formula: C_11_H_16_N_3_O_14_P_3_; MW: 511.21 g/mol; ^1^H NMR (700 MHz, D_2_O, signals for triethylammonium denoted by *) δ 8.14 (d, *J* = 7.8 Hz, 1H), 6.25 (d, *J* = 9.5 Hz, 1H), 6.24 (brs, 1H), 4.37−4.33 (m, 1H), 4.29−4.20 (m, 3H), 3.84 (d, *J* = 11.5 Hz, 1H), 3.72 (d, *J* = 11.5 Hz, 1H), 3.20 (q, *J* = 7.3 Hz, 24H*), 1.28 (t, *J* = 7.3 Hz, 36H*), 0.88 (s, 3H) ppm, OH and NH_2_ signals are missing possibly due to exchange in D_2_O; ^13^C NMR (176 MHz, D_2_O, signals for triethylammonium denoted by *) δ 163.1, 153.8, 143.4, 95.8, 89.3, 81.8 (d, *J_c-p_* = 8.9 Hz), 73.7, 64.0, 63.4 (d, *J_c-p_* = 5.0 Hz), 49.6, 46.6*, 16.0, 8.2* ppm; ^31^P NMR (162 MHz, D_2_O) −10.88 (d, *J* = 10.6 Hz, 1P), −11.34 (d, *J* = 19.6 Hz, 1P), −23.24 (brs, 1P) ppm; HRMS (ESI) *m/z* [M − H]⁻ calcd for C_11_H_19_N_3_O_14_P_3_:510.0085; found 510.0081 (−0.87 ppm) and [M + Na − H]⁻ calcd for C_11_H_18_N_3_NaO_14_P_3_: 531.9905; found 531.9900 (−0.88 ppm).

*((2R,3S,4R,5R)-5-(6-Amino-9H-purin-9-yl)-3-hydroxy-4-(hydroxymethyl)-4-methyltetrahydrofuran-2-yl)methyl tetrahydrogen triphosphate:* **6** (LCB-2344). Following general procedure C, (Bu_3_HN)_2_H_2_P_2_O_7_ (51.0 mg, 0.092 mmol, 1.30 equiv.), NBu_3_ (0.28 mL, 0.25 M), SalPCl (17.0 mg, 0.085 mmol, 1.20 equiv.) and nucleoside analogue **46** (21.0 mg, 0.071 mmol, 1.00 equiv.) in anhydrous DMF (0.7 mL, 0.1 M) were employed in the phosphorylation. Purification by reverse phase C18 flash chromatography (flow rate of 8 mL/min., gradient run of acetonitrile from 0 to 8% in 20 mM TEAAc, pH =7) provided nucleoside triphosphate **6** (LCB-2344) triethylammonium salt (3.4 mg, 5%) as a white powder. Formula: C_12_H_20_N_5_O_13_P_3_; MW: 535.23 g/mol; ^1^H NMR (500 MHz, D_2_O, signals for triethylammonium denoted by *) δ 8.60 (Br s, 1H), 8.28 (s, 1H), 6.43 (s, 1H), 4.50 (Br d, *J* = 8.6 Hz, 1H), 4.43–4.30 (m, 3H), 3.93 (d, *J* = 11.9 Hz, 1H), 3.83 (d, *J* = 11.6 Hz, 1H), 3.21 (q, *J* = 7.3 Hz, 21H*), 1.29 (t, *J* = 7.3 Hz, 30H*), 0.72 (s, 3H) ppm; OH and NH_2_ signals are missing possibly due to exchange in D_2_O; ^13^C NMR (176 MHz, D_2_O, signals for triethylammonium denoted by *) δ 152.8, 148.7, 148.4, 118.2, 88.4, 82.11 (d, *J* = 7.5 Hz), 73.7, 63.9, 63.4, 50.0, 46.6*, 15.7, 8.2* ppm. One aromatic carbon signal could not be found, with the small amount of sample to isolate (J values result from ^13^C−^31^P coupling and were assigned when possible); ^31^P NMR (162 MHz, D_2_O) δ −6.43 (d, *J* = 21.2 Hz, 1P), −11.35 (d, *J* = 19.9 Hz, 1P), −22.54 (t, *J* = 20.6 Hz, 1P) ppm; HRMS (ESI) *m/z* [M − H]^−^ calcd for C_12_H_19_N_5_O_13_P_3_: 534.0198; found 534.0197 (−0.08 ppm).

*((2R,3S,4R,5R)-5-(6-Amino-2-chloro-9H-purin-9-yl)-3-hydroxy-4-(hydroxymethyl)-4-methyltetrahydrofuran-2-yl)methyl triphosphate:* **7** (LCB-2279). Following general procedure C, (Bu_3_HN)_2_H_2_P_2_O_7_ (100 mg, 0.182 mmol, 2.00 equiv.), NBu_3_ (0.36 mL, 0.25 M), SalPCl (37.0 mg, 0.182 mmol, 2.00 equiv.) and nucleoside analogue **45** (30.0 mg, 0.091 mmol, 1.00 equiv.) in anhydrous DMF (1 mL, 0.1 M) were employed in the phosphorylation. Purification by reverse phase C18 flash chromatography (flow rate of 12 mL/min., gradient run of acetonitrile from 0 to 20% in 20 mM TEAAc, pH = 7) provided nucleoside triphosphate **7** (LCB-2279) triethylammonium salt (10 mg, 11%) as a white powder. Formula: C_12_H_19_ClN_5_O_13_P_3_; MW: 569.68 g/mol; ^1^H NMR (500 MHz, D_2_O, signals for triethylammonium denoted by *) δ 8.52 (Br s, 1H), 6.29 (s, 1H), 4.48 (d, *J* = 8.4 Hz, 1H), 4.41–4.26 (m, 3H), 3.91 (d, *J* = 11.7 Hz, 1H), 3.82 (d, *J* = 11.6 Hz, 1H), 3.19 (q, *J* = 6.2 Hz, 20H*), 1.27 (t, *J* = 7.3 Hz, 30H*), 0.72 (s, 3H) ppm; OH and NH_2_ signals are missing possibly due to exchange in D_2_O; ^13^C NMR (176 MHz, D_2_O, signals for triethylammonium denoted by *) δ 156.3, 153.7, 149.7, 140.9, 117.3, 88.1, 82.1 (d, *J_c-p_* = 8.8 Hz), 73.8, 63.9 (d, *J_c-p_* = 5.3 Hz), 63.5, 49.9, 46.6*, 15.7, 8.2* ppm (J values result from ^13^C−^31^P coupling and were assigned when possible); ^31^P NMR (162 MHz, D_2_O) −6.42 (d, *J* = 21.5 Hz, 1P), −11.36 (d, *J* = 19.9 Hz, 1P), −22.50 (t, *J* = 20.6 Hz, 1P) ppm; HRMS (ESI) *m/z* [M − H]^−^ calcd for C_12_H_18_ClN_5_O_13_P_3_: 567.9808; found 567.9819 (+1.86 ppm) and [M + Na − H] calcd for C_12_H_17_ClN_5_NaO_13_P_3_: 589.9627; found 589.9637 (+1.62 ppm). A minor and inseparable impurity was isolated with the compound. ^31^P NMR signals at −11.04 (d) and −6.49 (d) ppm suggest that this side product could be the corresponding diphosphate.

### 3.3. Biology

SARS-CoV-2 RdRp complex (nsp12/nsp7/nsp8) was prepared as published [9]. The RNA synthesis activity of the RdRp complex was evaluated in a reaction mixture comprising a 19-mer RNA primer, 43-mer RNA template, 25 mM TRIS-HCl (pH8), cold NTPs (50 µM ATP, CTP and UTP; 25 µM GTP), 0.1 μM [α-32P]-GTP and different concentrations of each inhibitor of interest. Nuclease-free water was added in place of the RdRp or the inhibitors for the negative control (−Pol) or no-treatment control (+Pol), respectively. After 10 min incubation at 30 °C, 5mM MnCl2 was added into each reaction mix to initiate the RdRp reaction. After another 30 min incubation at 30 °C, the RdRp reactions were terminated with formamide containing 40 mM EDTA, and were heated at 95 °C for 10 min. The resulting reaction products were resolved on 20% polyacrylamide-urea denaturing gels (SequaGel, National Diagnostics, Atlanta, GA, USA) and visualized using the Amersham Typhoon IP (Cytiva, Marlborough, MA, USA). Analyses were subsequently conducted with ImageQuant TL 8.2 (Cytiva) [6].

## 4. Conclusions

The syntheses of novel NTP analogues bearing a quaternary all-carbon stereogenic center at C3′ and C2′ have been achieved. The stereogenic quaternary center at C2′ or C3′ were generated by photocatalyzed cyclization/elimination free-radical-based reactions through five-exo-trig cyclization in the C2′ quaternary series and seven-endo trig cyclization in the C3′ quaternary series. The installation of the hydroxyl at C2′ was significantly improved through a stereoselective epoxidation, providing access to NA-containing quaternary carbon at C3′. A modified approach was also presented for the synthesis of NAs of C2′ quaternary center series. The synthesis and purification of the corresponding nucleoside triphosphates are reported for the first time. Optimization for NTPs bearing C2′ quaternary stereogenic center is under development.

Finding novel molecules that could act as antiviral agents against SARS-CoV-2, or other emerging viruses, is an important venue for the present and future treatment of these infections. We have reported herein two inhibitors, **6** (LCB-2344) and **7** (LCB-2279), against SARS-CoV-2 RdRp, which are lead molecules for further optimization. The stereogenic quaternary center at C2′ will be further modified to improve the potency of the novel series of molecules. Currently the study is underway to explore the monophosphorylated pro-drugs of these molecules and their antiviral efficacity.

## Data Availability

The data presented in this study are available in the paper and Appendix A.

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
