# Peer review of "Nucleotide Analogues Bearing a C2′ or C3′-Stereogenic All-Carbon Quaternary Center as SARS-CoV-2 RdRp Inhibitors†"

_molecules, 2022, doi:10.3390/molecules27020564_

Round 1

Reviewer 1 Report

This manuscript investigated different synthetic routes for obtaining ribofuranoside fragments for compounds with potential inhibition on SARS-CoV-2. The authors clearly express the synthetic descriptions, which strengthens the discussion of the work. Furthermore, the presentation of the methodologies are adequate, and the results are precise. Therefore, I have only a few suggestions:

-    Add 2D spectra (NOESY, HMBC and HSQC) in the supplementary material.
-    Add DEPT-135 spectra to strengthen the evidence for the distinction of quaternary and tertiary carbons in the synthesised structures' 2' and 3' positions.

Author Response

We greatly appreciated your comments on our article. As suggested, we have added the 2D NMR spectra for the following compounds:

(i) NOESY spectra for compounds 26-29.

(ii) NOESY, HMBC and HSQC spectra for compounds 30-31, 33-34 and 44.

We have not recorded the DEPT-135 experiments for these compounds.

Reviewer 2 Report

  • The abstract should contain the main important results
  • In the introduction the importance of heterocyclic bases containing nitrogen atom must be involved based on the following references:

https://www.mdpi.com/1420-3049/23/10/2548

https://www.sciencedirect.com/science/article/abs/pii/S0022286021018536

https://www.ingentaconnect.com/contentone/ben/mrmc/2021/00000021/00000001/art00012

  • The structural activity relationships studies must be added
  • References must be updated (1990)

Author Response

We greatly appreciated your comments regarding the introduction and structural activity relationship of our molecules. In this publication, we mainly focused on the NTPs containing naturally occurring nitrogenous nucleobases, with the exception of 2-chloroadenine. Hence, we have not included a description of modified nucleobases activity.

            As requested, the abstract has been revised and we have added a recent reference (24) for the epoxidation reaction in addition to the previous reference (23). We are currently making a small library of NAs by installing modified nucleobases and functionalization at hydroxymethyl of quaternary stereocenters to examine their activity against RdRp. This work will allow us to better comprehend the structure activity relationships and will be published in due course.

Reviewer 3 Report

This paper describe study of synthesis of nucleotides having stereogenic centers at C2’ / C3’ bearing adenine or cytosine nucleobases and their inhibitory profile against RNA-dependent RNA polymerase (RdRp).

All the significant studies such were made.

Also preliminary (also if limited) study for SARS-CoV-2 RNA-dependent RNA polymerase inhibition was performed to demonstrate their conclusion. 
Docking studies on RdRp could be added in supporting information to study possible interaction of two active compounds and compare to Remdesivir? 

Hence, this reviewer indicate accept this MS for publication after minor revision in Molecules.   

Author Response

We greatly appreciated your comments on our article. We have not performed docking studies with our NAs and RdRp in the context of our pilot studies. We are currently focusing on the modification at nucleobases and functionalization at hydroxymethyl of quaternary stereocenter for our NAs to improve their activity against RdRp. Docking studies will be performed in parallel to guide these modifications. 

Round 2

Reviewer 2 Report

accepted